# Watermarking Graph Neural Networks via Explanations for Ownership Protection

Jane Downer [1] [*]  Yingdan Shi [1] [*]  Ziyan Liu [2] [*] [†]  Ren Wang [1]  Binghui Wang [1]

## Abstract

Graph Neural Networks (GNNs) are widely deployed in industry, making their intellectual property valuable. However, protecting GNNs from unauthorized use remains a challenge. Watermarking offers a solution by embedding ownership information into models. Existing watermarking methods have two limitations: First, they rarely focus on graph data or GNNs. Second, the *de facto* backdoor-based method relies on manipulating training data, which can introduce ownership ambiguity through misclassification and vulnerability to data poisoning attacks that can interrupt the backdoor mechanism. Our explanation-based watermarking inherits the strengths of backdoor-based methods (e.g., black-box verification) without data manipulation, eliminating ownership ambiguity and data dependencies. In particular, we watermark GNN explanations such that these explanations are statistically distinct from others, so ownership claims must be verified through statistical significance. We theoretically prove that, even with full knowledge of our method, locating the watermark is NP-hard. Empirically, our method demonstrates robustness to fine-tuning and pruning attacks. By addressing these challenges, our approach significantly advances GNN intellectual property protection.

## 1. Introduction

Graph Neural Networks (GNNs) (Scarselli et al., 2008; Kipf & Welling, 2017; Hamilton et al., 2018; Veličković et al., 2018) are widely used for graph-structured data tasks, such as social network analysis, bioinformatics, and recommendation systems (Zhang et al., 2021; Wang et al., 2021b; Zhou et al., 2020; Islam et al., 2026; Zhao et al., 2026; Zhang et al., 2025; Wu et al., 2025). Various giant companies have depend on GNNs: Amazon for product recommendations (Virinchi, 2022); Google's TensorflowGNN for Maps traffic prediction (Sibon Li et al., 2021; Lange & Perez, 2020); Meta for friend/content recommendations (MetaAI, 2023); Alibaba's AliGraph for fraud and risk detection (Yang, 2019; Liu et al., 2021b). Given significant GNN development, ownership verification is crucial to protect against illegal copying, model theft, and malicious distribution.

Watermarking embeds secret patterns into models (Uchida et al., 2017) to verify ownership. As a *de facto* approach, backdoor-based watermarking (Adi et al., 2018; Bansal et al., 2022; Lv et al., 2023; Yan et al., 2023; Li et al., 2022; Shao et al., 2022; Lansari et al., 2023; Yang et al., 2024b) insert the watermark pattern as a "trigger" into clean samples with altered *target labels*, and trains on both watermarked and clean data. During verification, ownership is demonstrated by producing the triggered samples that yield the target label. Backdoor-based watermarking methods have several merits: they are robust to removal attacks (pruning and fine-tuning), and verification only requires black-box model access.

However, recent works (Yan et al., 2023; Liu et al., 2024; Xu et al., 2023) reveal a fundamental limitation: backdoor-based methods induce ownership ambiguity, as attackers could falsely claim misclassified data as ownership evidence. Additionally, embedding watermarks into data properties is vulnerable to data poisoning attacks, where an adversary can manipulate the data to disrupt the watermarking process (Steinhardt et al., 2017; Zhang et al., 2019; Downer et al., 2025). Recognizing these limitations, researchers have explored alternate watermark embedding spaces. Shao et al. (2024) embed watermarks DNN prediction *explanations*, avoiding tampering with predictions or parameters. While offering a compelling alternative to backdoor-based watermarking, their approach assumes a known ground-truth watermark, introducing challenges like a third-party verification requirement and potential disputes over the true watermark. Moreover, they do not address graph data's

---

[*]Equal contribution . [†]Work conducted through a research collaboration under the supervision of Prof. Ren Wang, the Trustworthy and Intelligent Machine Learning Research Lab, Illinois Institute of Technology. [1]Illinois Institute of Technology [2]Harrisburg University of Science and Technology. Correspondence to: Binghui Wang <bwang70@illinoistech.edu>, Ren Wang <rwang74@iit.edu>.

*Proceedings of the 43rd International Conference on Machine Learning*, Seoul, South Korea. PMLR 306, 2026. Copyright 2026 by the author(s).

unique complexities, including structural dependencies and multi-hop relationships.

We extend explanation-based watermarks to GNNs, additionally addressing graph-specific challenges and avoiding the need of a ground-truth watermark for verification. Our approach aligns explanations of selected subgraphs with a predefined watermark, ensuring robustness to removal attacks and preserving advantages of explanation-based methods. In doing so, we present the first explanation-based watermarking method tailored to GNNs.

**Our approach:** We develop a novel watermarking strategy for protecting GNN model ownership that both inherits the merits from and mitigates the drawbacks of backdoor-based watermarking. Like backdoor-based methods, our approach only needs black-box model access. However, in contrast to using *predictions* on the *polluted* watermarked samples, we leverage the *explanations* of GNN predictions on *clean* samples and align them with a predefined watermark for ownership verification.

Before training, the owner selects secret watermarked subgraphs (private) and defines a watermark pattern (*possibly* private).[1] The GNN trains with a dual-objective loss function that minimizes (1) classification loss, and (2) distance between the watermark and watermarked subgraph explanations. Like GraphLIME (Huang et al., 2023), we use Gaussian kernel matrices to approximate node feature influence on predictions. However, instead of an iterative approach, we use ridge regression to compute feature attribution vectors in a single step, providing a more efficient, closed-form solution.

Our approach is (i) *Effective*: Explanations of watermarked subgraphs exhibit high similarity to the watermark after training. (ii) *Unique:* This similarity across explanations is statistically unlikely without watermarking, and hence serves as our ownership evidence. (iii) *Undetectable:* We prove that, even with full knowledge of our watermarking method, finding the private watermarked subgraphs is computationally intractable (NP-hard). (iv) *Robust:* Empirical evaluations on multiple benchmark graph datasets and GNN models demonstrate robustness to fine-tuning and pruning-based watermark removal attacks. We summarize our contributions as follows:

- We introduce the first known method for watermarking GNNs via their explanations, eliminating ownership ambiguity and avoiding data manipulation problems of black-box watermarking schemes.

- We prove that it is NP-hard for the worst-case adversary to identify our watermarking mechanism.

---

[1]Ownership verification does not require knowledge of the watermark pattern.

- We show our method is robust to watermark removal attacks like fine-tuning and pruning.

## 2. Related Work

**White-Box Watermarking.** White-box watermarking techniques (Darvish Rouhani et al., 2019; Uchida et al., 2017; Wang & Kerschbaum, 2020; Shafieinejad et al., 2021) directly embed watermarks into the model parameters or features during training. For example, (Uchida et al., 2017) embed a watermark via a regularization term, while (Darvish Rouhani et al., 2019) propose embedding the watermark into the activation/feature maps. Although these methods are robust in theory (Chen et al., 2022), they require full access to the model parameters during verification, which may not be feasible in real-world scenarios, especially for deployed models operating in black-box environments (e.g., APIs).

**Black-Box Watermarking.** Black-box approaches verify model ownership using only model predictions (Adi et al., 2018; Chen et al., 2018b; Szyller et al., 2021; Le Merrer et al., 2019). They often use backdoor mechanisms, training models to output specific predictions for "trigger" inputs as ownership evidence (Adi et al., 2018; Zhang et al., 2018; Yang et al., 2024b). These methods have significant downsides. First, watermarks embedded into data features can be interrupted by data poisoning attacks (Steinhardt et al., 2017; Zhang et al., 2019). Further, backdoor methods suffer from ambiguity — attackers may claim naturally-misclassified samples as their own "watermark" (Yan et al., 2023; Liu et al., 2024). Given these issues with backdoor-based methods, (Shao et al., 2024) proposed embedding DNN watermarks in explanations to avoid prediction manipulation and maintain black-box compatibility.

**Watermarking GNNs.** Varying size and structure of graphs make watermark embedding challenging. Moreover, GNNs' multi-hop message-passing mechanisms are more sensitive to data changes than neural networks processing more uniform data like images or text (Wang & Gong, 2019; Zügner et al., 2020; Mu et al., 2021; Wang et al., 2021a; 2022; Zhou et al., 2023; Wang et al., 2023; 2024; Yang et al., 2024a; Zhao et al., 2025; Li & Wang, 2025; Li et al., 2025). The only existing black-box watermarking GNNs (Xu et al., 2023) suffer from the same issue as backdoor watermarking of non-graph models (Liu et al., 2024) [2]. These issues, coupled with graph complexity, make existing watermarking techniques unsuitable for GNNs. This highlights the need for novel schemes.

---

[2]Fingerprinting method (Waheed et al., 2024) verifies GNN ownership with node embeddings instead of explicit watermark patterns. However, it is vulnerable to pruning attacks. Relying on intrinsic model features can limit uniqueness guarantees and risk ownership ambiguity (Wang et al., 2021d; Liu et al., 2024).

# 3. Background and Problem Formulation

## 3.1. GNNs for Node Classification

Let a graph be denoted as $G = (\mathcal{V}, \mathcal{E}, \mathbf{X})$, where $\mathcal{V}$ is the set of nodes, $\mathcal{E}$ is the set of edges, and $\mathbf{X} = [\mathbf{x}_1, \cdots, \mathbf{x}_N] \in \mathbb{R}^{N \times F}$ is the node feature matrix. $N = |\mathcal{V}|$ is the number of nodes, $F$ is the number of features per node, and $\mathbf{x}_u \in \mathbb{R}^F$ is the node $u$'s feature vector. We assume the task of interest is node classification. In this context, each node $v \in \mathcal{V}$ has a label $y_v$ from a label set $C = \{1, 2, \cdots, C\}$, and we have a set of $|\mathcal{V}^{tr}|$ labeled nodes $(\mathcal{V}^{tr}, \mathbf{y}^{tr}) = \{(v_u^{tr}, y_u^{tr})\}_{u \in \mathcal{V}^{tr}} \subset \mathcal{V} \times C$ nodes as the training set. A GNN for node classification takes as input the graph $G$ and training nodes $\mathcal{V}^{tr}$, and learns a node classifier, denoted as $f$, that predicts the label $\hat{y}_v$ for each node $v$. Suppose a GNN has $L$ layers and a node $v$'s representation in the $l$-th layer is $\mathbf{h}_v^{(l)}$, where $\mathbf{h}_v^{(0)} = \mathbf{x}_v$. Then it updates $\mathbf{h}_v^{(l)}$ for each node $v$ using the following two operations:

$$\mathbf{l}_v^{(l)} = \texttt{Agg}(\{\mathbf{h}_u^{(l-1)} : u \in \mathcal{N}(v)\}), \mathbf{h}_v^{(l)} = \texttt{Comb}(\mathbf{h}_v^{(l-1)}, \mathbf{l}_v^{(l)}), \quad (1)$$

where $\texttt{Agg}$ aggregates representations of a node's neighbors, and $\texttt{Comb}$ combines a node's previous representation and aggregated representation of that aggregation to update the node representation. $\mathcal{N}(v)$ denotes the neighbors of $v$. Different GNNs use different $\texttt{Agg}$ and $\texttt{Comb}$ operations.

The last-layer representation $\mathbf{h}_v^{(L)} \in \mathbb{R}^{|C|}$ of training nodes $v \in \mathcal{V}^{tr}$ are used to train the node classifier $f$. Let $\Theta$ be the model parameters and $v$'s softmax scores be $\mathbf{p}_v = f_\Theta(\mathcal{V}^{tr})_v = \text{softmax}(\mathbf{h}_v^{(L)})$, where $p_{v,c}$ is the probability of $v$ being class $c$. $\Theta$ are learned by minimizing a classification (e.g., cross-entropy) loss on the training nodes:

$$\Theta^* = \arg\min_\Theta \mathcal{L}_{CE}(\mathbf{y}^{tr}, f_\Theta(\mathcal{V}^{tr})) = -\Sigma_{v \in \mathcal{V}^{tr}} \ln p_{v,y_v}. \quad (2)$$

## 3.2. GNN Explanation

GNN explanations identify graph features that influence predictions. Some methods (e.g., GNNExplainer (Ying et al., 2019) and PGExplainer (Luo et al., 2020)) identify important subgraphs, while others (e.g., GraphLime (Huang et al., 2023)) identify key node features. Inspired by GraphLime (Huang et al., 2023), we use Gaussian kernel matrices to capture relationships between node features and predictions: Gaussian kernel matrices effectively capture nonlinear dependencies and complex variable relationships, ensuring subtle patterns in the data are effectively represented (Yamada et al., 2012). Using these Gaussian kernel matrices, we employ a closed-form solution with ridge regression (Hoerl & Kennard, 1970) to compute feature importance in a single step.

Our function $explain(\cdot)$ takes node feature matrix $\mathbf{X}$ and softmax scores $\mathbf{P} = [\mathbf{p}_1, \cdots, \mathbf{p}_N]$, yielding $F$-dimensional attribution vector $\mathbf{e}$ showing each feature's influence on predictions across nodes. This computes feature attributions ($\mathbf{e}$) by leveraging the relationships between input features ($\mathbf{X}$) and output predictions ($\mathbf{P}$) through Gaussian kernel matrices.

$$\mathbf{e} = explain(\mathbf{X}, \mathbf{P}) = (\tilde{K}^T \tilde{K} + \lambda I_F)^{-1} \tilde{K}^T \tilde{L}. \quad (3)$$

We defer precise mathematical definitions to Appendix B. For high-level understanding, the matrix $\tilde{K}$ ($N^2 \times F$) encodes pairwise feature similarities between nodes via a Gaussian kernel. $\tilde{L}$ ($N^2 \times 1$) uses a Gaussian kernel to encode pairwise prediction similarities between nodes. The term $(\tilde{K}^T \tilde{K} + \lambda I_F)^{-1}$, where $\lambda$ is a regularization hyperparameter and $I_F$ is the $F \times F$ identity matrix, solves a ridge regression problem to ensure a stable and interpretable solution. The product $\tilde{K}^T \tilde{L}$ ($F \times 1$) ties the Gaussian feature similarities ($\tilde{K}$) to the output prediction similarities ($\tilde{L}$), ultimately yielding the vector $\mathbf{e}$ ($F \times 1$), which quantifies the importance of each input feature for the GNN's predictions. This kernel-based method allows us to (1) aggregate node-level predictions into subgraph-level information; (2) utilize GNN-predicted probabilities over all categories; and (3) compared to GNNExplainer (Ying et al., 2019), keep the watermark subgraph private.

In this paper, the *explanation* of a GNN's node predictions means this feature attribution vector $\mathbf{e}$.

## 3.3. Problem Formulation

We propose an explanation-based GNN watermarking method. Our approach defines a watermark pattern ($\mathbf{w}$) and selects subgraphs from $G$. The GNN $f$ is trained to embed the relationship between $\mathbf{w}$ and these subgraphs, enabling their explanations to act as verifiable ownership evidence.

**Threat Model:** There are three parties: the model owner, the adversary, and the third-party model ownership verifier. Obviously, the model owner has white-box access to the target GNN model.

- **Adversary:** We investigate an adversary who falsely claims to own GNN model $f$. We assume they lacks knowledge of the watermarked subgraphs in $G$, but we also evaluate robustness under challenging scenarios where they might know specific details (e.g., shape or number of watermarked subgraphs). The adversary tries to undermine the watermark by (1) searching for the watermarked subgraphs (or similarly-convincing alternatives), or (2) implementing a removal attack.

- **Model Ownership Verifier:** Following existing backdoor-based watermarking, we use black-box ownership verification, where the verifier does not need full access to the protected model.

**Objectives:** Our explanation-based watermarking method aims to achieve the below objectives:

1. **Effectiveness.** Training must embed the watermark in the explanations of our selected subgraphs: their feature attribution vectors must be *sufficiently*[3] aligned with vector $\mathbf{w}$.

2. **Uniqueness.** Aligning watermarked subgraph explanations with $\mathbf{w}$ must yield statistically-significant similarity between explanations that is unlikely to occur in alternate solutions.

3. **Robustness.** The watermark must be robust to removal attacks like fine-tuning and pruning.

4. **Undetectability.** Non-owners should be unable to locate the watermarked explanations.

## 4. Methodology

Our watermarking method has three stages: (1) design, (2) embedding, and (3) ownership verification. We introduce stages (2) and (3) first as design relies on them.

Training $f$ uses a dual-objective loss function balancing node classification and watermark embedding. Minimizing watermark loss aligns $\mathbf{w}$ with explanations of $f$'s predictions on watermarked subgraphs, embedding the watermark. Verification tests for explanations statistically-significant similarity from their common alignment with $\mathbf{w}$. Lastly, we detail a watermark design that ensures this statistical significance, which provides unambiguous ownership evidence. Figure 1 overviews our method.

### 4.1. Watermark Embedding

Let training set $\mathcal{V}^{tr}$ be split as two disjoint subsets: $\mathcal{V}^{clf}$ for node classification and $\mathcal{V}^{wmk}$ for watermarking. Select $T$ subgraphs $\{G_1^{wmk}, \ldots, G_T^{wmk}\}$ whose nodes $\{\mathcal{V}_i^{wmk}\}_{i=1}^T$ will be watermarked. These subgraphs have explanations $\{\mathbf{e}_1^{wmk}, \ldots, \mathbf{e}_T^{wmk}\}$, where $\mathbf{e}_i^{wmk} = explain(\mathbf{X}_i^{wmk}, \mathbf{P}_i^{wmk})$ explains $f$'s softmax output $\mathbf{P}_i^{wmk}$ on $G_i^{wmk}$'s nodes $\mathcal{V}_i^{wmk}$, with features $\mathbf{X}_i^{wmk}$. Define watermark $\mathbf{w}$ as an $M$-dimensional vector ($M \leq F$), with entries of 1s and $-1$s.

Inspired by (Shao et al., 2024), we use multi-objective optimization to balance classification performance with a hinge-like *watermark loss*. Minimizing this loss encourages alignment between $\mathbf{w}$ and $\{\mathbf{e}_i^{wmk}\}_{i=1}^T$, embedding the relationship between $\mathbf{w}$ and these subgraphs.

$$\mathcal{L}_{wmk}(\{\mathbf{e}_i^{wmk}\}_{i=1}^T, \mathbf{w}) = \Sigma_{i=1}^T \Sigma_{j=1}^M \max(0, \epsilon - \mathbf{w}[j] \cdot \mathbf{e}_i^{wmk}[\mathbf{idx}[j]]), \quad (4)$$

where $\mathbf{e}_i^{wmk}[\mathbf{idx}]$ represents the *watermarked portion* of $\mathbf{e}_i^{wmk}$ on node feature indices $\mathbf{idx}$ with length $M$; $\mathbf{idx}$ is

same for all explanations $\{\mathbf{e}_i^{wmk}\}_{i=1}^T$. We emphasize that $\mathbf{idx}$ are not arbitrary, but are rather the result of design choices discussed later in Section 4.3. The hyperparameter $\epsilon$ bounds the contribution of each multiplied pair $\mathbf{w}[j] \cdot \mathbf{e}_i^{wmk}[\mathbf{idx}[j]]$ to the summation.

We train the GNN model $f$ to minimize both classification loss on the nodes $\mathcal{V}^{clf}$ (see Equation 2) and watermark loss on the explanations of $\{G_1^{wmk}, \ldots, G_T^{wmk}\}$, with a balancing hyperparameter $r$:

$$\min_{\Theta} \mathcal{L}_{CE}(\mathbf{y}^{clf}, f_{\Theta}(\mathcal{V}^{clf})) + r \cdot \mathcal{L}_{wmk}(\{\mathbf{e}_i^{wmk}\}_{i=1}^T, \mathbf{w}). \quad (5)$$

After training, the learned parameters $\Theta$ ensures not only an accurate node classifier, but also similarity between $\mathbf{w}$ and explanations $\{\mathbf{e}_i^{wmk}\}_{i=1}^T$ at indices $\mathbf{idx}$. See Algorithm 1 in Appendix for the details.

### 4.2. Ownership Verification

Since they were aligned with the same $\mathbf{w}$, explanations $\{\mathbf{e}_i^{cdt}\}_{i=1}^T$ will be similar to each other after training. Therefore, when presented with $T$ *candidate subgraphs* $\{\mathbf{e}_1^{cdt}, \mathbf{e}_2^{cdt}, \cdots, \mathbf{e}_T^{cdt}\}$ by a purported owner (note that our threat model assumes a strong adversary who also knows $T$), we must measure the similarity between these explanations to verify ownership. If the similarity is statistically significant at a certain level, we can conclude the purported owner knows which subgraphs were watermarked during training, and therefore that they are the true owner.

**Explanation Matching:** Our GNN explainer in Equation (3) gives a positive or negative score for each node feature, indicating its influence on the GNN's predictions, generalized across all nodes in the graph. To easily compare these values across candidate explanations, we first *binarize* them with the sign function. For the $j^{th}$ index of explanation $\mathbf{e}_i^{cdt}$, this process is defined as:

$$\hat{\mathbf{e}}_i^{cdt}[j] = \begin{cases} 1 & \text{if } \mathbf{e}_i^{cdt}[j] > 0, \\ -1 & \text{if } \mathbf{e}_i^{cdt}[j] < 0, \\ 0 & \text{otherwise.} \end{cases} \quad (6)$$

We then count the *matching indices* (MI) across all the binarized explanations — the number of indices at which all binarized explanations have matching, non-zero values:[4]

$$\begin{aligned} \text{MI}^{cdt} &= \text{MI}(\{\hat{\mathbf{e}}_i^{cdt}\}_{i=1}^T) \\ &= \Sigma_{j=1}^F \mathbb{1}(((\{\hat{\mathbf{e}}_i^{cdt}[j] \neq 0, \forall i\}) \wedge (\hat{\mathbf{e}}_1^{cdt}[j] = \cdots = \hat{\mathbf{e}}_T^{cdt}[j])). \end{aligned} \quad (7)$$

**Approximating a Baseline MI Distribution:** To test $\text{MI}^{cdt}$ significance, we first approximate the distribution

---

[3]Note: alignment between explanations and $\mathbf{w}$ is a tool for the owner to measure optimization success; for a watermark to function as ownership evidence, alignment must simply be "good enough" (See Section 5.2.1).

[4]We exclude 0s from our MI count. A 0 in the explanation indicates no dependence between features and predictions, which could only result from extreme (unlikely) optimization precision. These 0s likely reflect existing 0s in $\mathbf{X}$, so we conclude they are irrelevant as watermarking metrics.

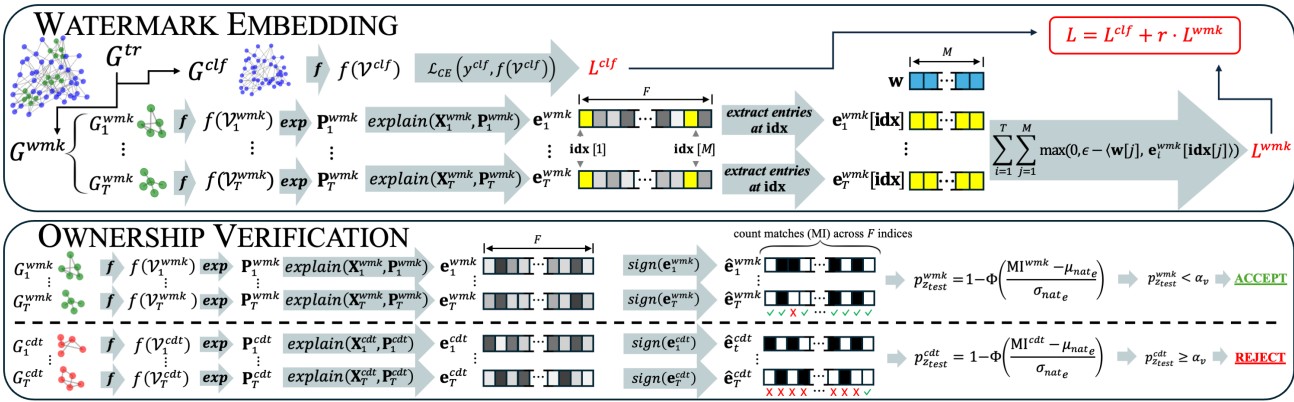

*Figure 1.* Overview: During embedding, $f$ is optimized to (1) minimize node classification loss and (2) align watermarked subgraph explanations with $\mathbf{w}$. The similarity of $G^{cdt}$'s binarized explanations, $\{\hat{\mathbf{e}}_i^{cdt}\}_{i=1}^{T}$, is tested for significance during ownership verification. In this example, $G^{cdt}$ are *not* the watermarked subgraphs; therefore, $\{\hat{\mathbf{e}}_i^{cdt}\}_{i=1}^{T}$ fail to exhibit significant similarity and are rejected.

of *naturally-occurring* matches: the MIs for all $T$-sized sets of un-watermarked explanations. This involves running $I$ simulations (sufficiently large; $I = 1000$ in our experiments), where we randomly sample sets of $T$ subgraphs from $G$ and compute the MI of the binarized explanations for each set. We then derive *empirical* estimates of the mean and standard deviation, $\mu_{nat_e}$ and $\sigma_{nat_e}$ (indicated by the subscript "e"), for the $I$ MIs.

**Significance Testing to Verify Ownership:** We verify the purported owner's ownership by testing if $\text{MI}^{cdt}$ is statistically unlikely for randomly selected subgraphs, at a significance level $\alpha_v$:

$$Ownership = \begin{cases} True & \text{if } p_{z_{test}} < \alpha_v, \\ False & \text{otherwise.} \end{cases} \quad (8)$$

where $z_{test} = \frac{\text{MI}^{cdt} - \mu_{nat_e}}{\sigma_{nat_e}}$. Algorithm 2 (Appendix) details the ownership verification process.

### 4.3. Watermark Design

The watermark $\mathbf{w}$ is an $M$-dimensional vector with entries of 1 and $-1$. The size and location of $\mathbf{w}$ must allow us to *effectively* embed *unique* ownership evidence into GNN.

**Design Goal:** The watermark should be designed to yield a *target MI* ($\text{MI}^{tgt}$) that passes the statistical test in Equation (8). This value is essentially the upper bound on a one-sided confidence interval. However, since we cannot get the estimates $\mu_{nat_e}$ or $\sigma_{nat_e}$ without a trained model, we instead use a binomial distribution to *predict* estimates $\mu_{nat_p}$ and $\sigma_{nat_p}$ (note the subscript "p").

We assume the random case, where a binarized explanation includes values $-1$ or $1$ with equal probability (again, ignoring zeros; see Footnote 4). Across $T$ binarized explanations, the probability of a match at an index is $p_{match} = 2 \times 0.5^T$. We estimate $\mu_{nat_p} = F \times p_{match}$ (where $F$ is number

of node features), and $\sigma_{nat_p} = \sqrt{F \times p_{match}(1 - p_{match})}$. We therefore define $\text{MI}^{tgt}$ as follows:

$$\text{MI}^{tgt} = min(\mu_{nat_p} + \sigma_{nat_p} \times z_{tgt}, F), \quad (9)$$

where $z_{tgt}$ is the $z$-score for target significance $\alpha_{tgt}$. In practice, we set $\alpha_{tgt} = 1e-5$; since $\text{MI}^{tgt}$ affects watermark design, we want to ensure it does not underestimate the upper bound.

**Watermark Length $M$:** For $T$ binarized explanations, our estimated lower bound of baseline MI is:

$$\text{MI}^{LB} = max(\mu_{nat_p} - \sigma_{nat_p} \times z_{LB}, 0), \quad (10)$$

where $z_{LB}$ is the $z$-score for target significance, $\alpha_{LB}$ — in practice, $\alpha_{LB}$ equals $\alpha_{tgt}$ ($1e - 5$).

We expect that our watermark must add ($\text{MI}^{tgt} - \text{MI}^{LB}$) net MI at most. However, natural matching between some indices in the binarized explanations may reduce the watermark's net contribution. We therefore pad watermark length. Padding is based on the probability of a natural match. In the worst case, where $\text{MI}^{tgt}$ indices naturally match, the probability of a watermarked index producing a new match is $(F - \text{MI}^{tgt})/F$. Consequently, we pad the required $M$ by the inverse, $F/(F - \text{MI}^{tgt})$:

$$M = \lceil (\text{MI}^{tgt} - \text{MI}^{LB}) \times F/(F - \text{MI}^{tgt}) \rceil. \quad (11)$$

Watermark length $M$ should yield enough net MI to reach the total, $\text{MI}^{tgt}$, that the owner needs to demonstrate ownership. Note that under the assumption that we set $\alpha_{LB}$ equal to $\alpha_{tgt}$, Equation (11) is ultimately a function of three variables: $\alpha_{tgt}$, $F$, and $T$.

**Watermark Location idx:** Each explanation corresponds to node feature indices. It is easiest to watermark indices at non-zero features. We advise selecting **idx** from the $M$

*Table 1.* Watermarking results. Each value is the average of five trials with distinct random seeds.

| | GCN | | SGC | | SAGE | | Transformer | |
|---|---|---|---|---|---|---|---|---|
| | Acc (Train/test) | | Acc (Train/test) | | Acc (Train/test) | | Acc (Train/test) | |
| **Dataset** | Wmk | No Wmk | Wmk | No Wmk | Wmk | No Wmk | Wmk | No Wmk |
| **Photo** | 91.3 / 89.4 | 90.9 / 88.3 | 91.4 / 89.9 | 90.1 / 88.0 | 94.2 / 90.8 | 94.1 / 88.2 | 99.9 / 90.7 | 95.0 / 86.8 |
| **PubMed** | 88.6 / 85.8 | 85.7 / 81.4 | 88.8 / 85.9 | 85.3 / 81.4 | 90.5 / 86.0 | 91.1 / 81.2 | 99.7 / 87.9 | 94.2 / 86.5 |
| **CS** | 98.5 / 90.3 | 96.8 / 89.8 | 98.4 / 90.3 | 96.7 / 90.1 | 100.0 / 88.4 | 99.9 / 88.9 | 100.0 / 93.1 | 99.4 / 92.2 |
| **Reddit2** | — | — | — | — | — | — | 83.4 / 79.4 | 81.0 / 80.4 |

| | Wmk Alignmt | MI $p$-val | Wmk Alignmt | MI $p$-val | Wmk Alignmt | MI $p$-val | Wmk Alignmt | MI ($p$-val) |
|---|---|---|---|---|---|---|---|---|
| **Photo** | 91.4 | <0.001 | 91.8 | <0.001 | 97.7 | <0.001 | 87.9 | <0.001 |
| **PubMed** | 91.5 | <0.001 | 88.9 | <0.001 | 85.2 | <0.001 | 94.0 | <0.001 |
| **CS** | 73.8 | <0.001 | 74.5 | <0.001 | 78.2 | <0.001 | 73.9 | <0.001 |
| **Reddit2** | — | — | — | — | — | — | 76.3 | <0.001 |

most frequently non-zero node features across all $T$ watermarked subgraphs. Let $\mathbf{X}^{wmk} = [\mathbf{X}_1^{wmk}; \mathbf{X}_2^{wmk}; \cdots \mathbf{X}_T^{wmk}]$ be the concatenation of node features of the $T$ watermarked subgraphs. Define **idx** as:

$$\mathbf{idx} = \text{top}_M\left(\{\|\mathbf{x}_1^{wmk}\|_0, \|\mathbf{x}_2^{wmk}\|_0, \cdots, \|\mathbf{x}_F^{wmk}\|_0\}\right), \quad (12)$$

where $\mathbf{x}_j^{wmk}$ is the $j$-th column of $\mathbf{X}^{wmk}$, $\|\cdot\|_0$ represents the number of non-zero entries in a vector, and $\text{top}_M(\cdot)$ returns the indices of the $M$ largest values.

### 4.4. Locating the Watermarked Subgraphs

An adversary may attempt to locate watermarked subgraphs to claim ownership. In the worst case, they have access to $G^{tr}$ and know $T$ (number of watermarked subgraphs) and $s$ (nodes per subgraph). With $G^{tr}$, they can compute the natural match distribution ($\mu_{nat_e}, \sigma_{nat_e}$) and search for $T$ subgraphs with maximally significant MI, using either brute-force or random search.

**Brute-Force Search:** If the training graph has $N$ nodes, identifying $n_{sub} = sN$-node subgraphs yields $\binom{N}{n_{sub}}$ options. To find the $T$ subgraphs with a maximum MI across their binarized explanations, an adversary must compare all $T$-sized sets of these subgraphs, with $\binom{\binom{N}{n_{sub}}}{T}$ sets in total.

Moreover, we can reduce the Maximum $k$-Subset Intersection (MSI) (Clifford & Popa, 2011) problem to the that of a brute search for $T$ effective subgraphs. MSI, known to be NP-hard, seeks the $k$ subsets with maximal intersection. Our reduction maps each MSI subset to a potential subgraph in $G$, with $k$ corresponding to our selected number of subgraphs, $T$. Finding $k$ sets with maximum intersection corresponds to finding $T$ subgraphs whose explanations share the maximum matching indices.

**Random Search:** Adversaries can search for a "good enough" group of subgraphs through random sampling, making $T$ random selections of $n_{sub}$-sized sets of nodes. Given $N$ training nodes and $T$ watermarked subgraphs of size $n_{sub}$,

the probability that an attacker-chosen subgraph of size $n_{sub}$ overlaps with any single watermarked subgraph with no less than $j$ nodes is given as:

$$P(\#\text{overlap-nodes} \geq j) = 1 - \left(\frac{\sum_{m=1}^{j} \binom{n_{sub}}{m}\binom{N-n_{sub}}{n_{sub}-m}}{\binom{N}{n_{sub}}}\right)^T. \quad (13)$$

The sum is the probability a randomly chosen subgraph and a watermarked subgraph share $< j$ nodes. Raising to power $T$ gives the probability all watermarked subgraphs have $< j$ overlap. 1 minus this value gives the probability the random subgraph and any watermarked subgraph share $\geq j$ nodes.

## 5. Experiments

### 5.1. Setup

**Datasets and Training/Testing Sets:** We evaluate our watermarking method on four node classification datasets: Amazon Photo (McAuley et al., 2015), CoAuthor CS (Shchur et al., 2019), PubMed (Yang et al., 2016), and Reddit2 (Zeng et al., 2019) (See Appendix A for details). *Our framework can also extend to other graph tasks; see Appendix G.* The graph is split into three sets: 60% for training, 20% for testing, and 20% for further training tasks (e.g., fine-tuning or other robustness evaluations). Training nodes divide into two disjoint sets: one for GNN classifier training, and one consisting of the watermarked subgraphs. (sizes as hyperparameters mentioned below.) The test set is for post-training classification evaluation. The remaining set enables additional training of the pre-trained GNN on unseen data to assess watermark robustness.

**GNN Models and Hyperparameters:** We apply our watermarking method to four GNN models: GCN (Kipf & Welling, 2017), SGC (Wu et al., 2019), SAGE (Hamilton et al., 2018), and Graph Transformer (Shi et al., 2020). Our main results use SAGE architecture, and $T = 4$ watermarked subgraphs, each with the size $s = 0.5\%$ of the

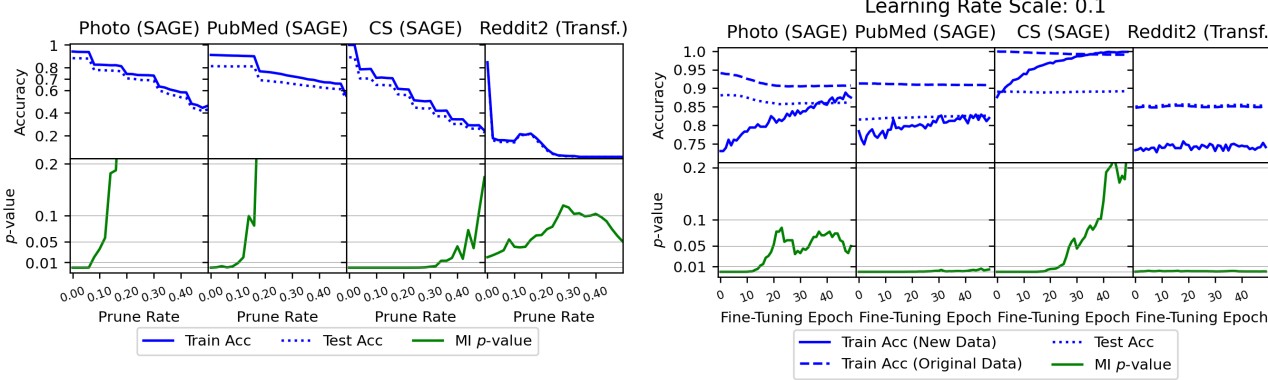

*Figure 2.* Effect of pruning (left) and fine-tuning (right) on MI *p*-value, under default settings (GraphSAGE, $T = 4$, $s = 0.005$). See Appendix figures 6-12 for results with varied settings.

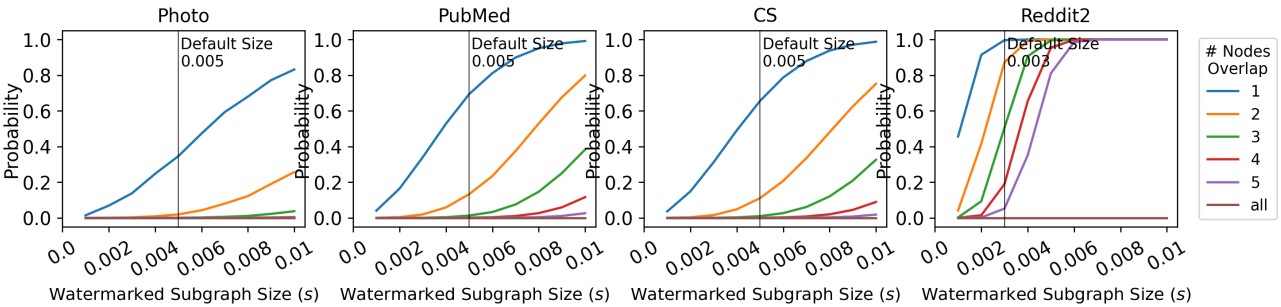

*Figure 3.* The probability that a randomly-chosen subgraph overlaps with a watermarked subgraph.

training nodes.[5] Key hyperparameters in our watermarking method, including the significance levels ($\alpha_{tgt}$ and $\alpha_v$), balanced hyperparameter ($r$), and watermark loss contribution bound ($\epsilon$), were tuned to balance classification and watermark losses. A list of all hyperparameter values are in the Appendix. *Note that our watermark design in Equation (11) allows us to learn the watermark length M.*

### 5.2. Results

As stated in Section 3.3, watermarks should be effective, unique, robust, and undetectable. Our experiments aim to assess each of these. (For more results see Appendix.)

#### 5.2.1. EFFECTIVENESS AND UNIQUENESS

Embedding *effectiveness* can be measured by the alignment of the binarized explanations with the watermark pattern **w** at indices **idx**; this metric can be used by the owner to confirm that **w** was effectively embedded in $f$ during training. Since the entries of **w** are 1s and $-1$s, we simply

---

[5]Reddit2 is by far the largest, with 232,965 nodes and 23,213,838 edges. For this reason, only Graph Transformer achieved convergence on Reddit2 in our experiments. Given Reddit's scale, we also default to the smaller $s = 0.03\%$ (46 nodes) for watermarked subgraph size.

count the average number of watermarked indices at which a binarized explanation matches **w**:

$$\begin{aligned}
&\textit{Watermark Alignment} \\
&= (1/T) \times \Sigma_{i=1}^{T} \Sigma_{j=1}^{M} \mathbb{1}(\hat{\mathbf{e}}_i^{wmk}[\mathbf{idx}[j]] = \mathbf{w}[j]).
\end{aligned} \quad (14)$$

Watermarking *uniqueness* is measured by the MI *p*-value for the binarized explanations of the $T$ watermarked subgraphs, as defined by Equation (8). A low *p*-value indicates the MI is statistically unlikely to be seen in explanations of randomly selected subgraphs. *If the watermarked subgraphs yield a uniquely large MI, it is sufficient, even if alignment is under 100%.*

Table 1 shows results under default settings, averaged over five trials with distinct random seeds and watermark patterns. The MI *p*-value is below 0.001 for all $T > 2$; this shows *uniqueness* of the ownership claim, meaning the embedding was sufficiently *effective*.

#### 5.2.2. ROBUSTNESS

We test our method against 4 different types of removal attacks: model pruning, fine-tuning, model merge and knowledge distillation attack. Here we present the results of pruning and fine-tuning. See Appendix E for more results of

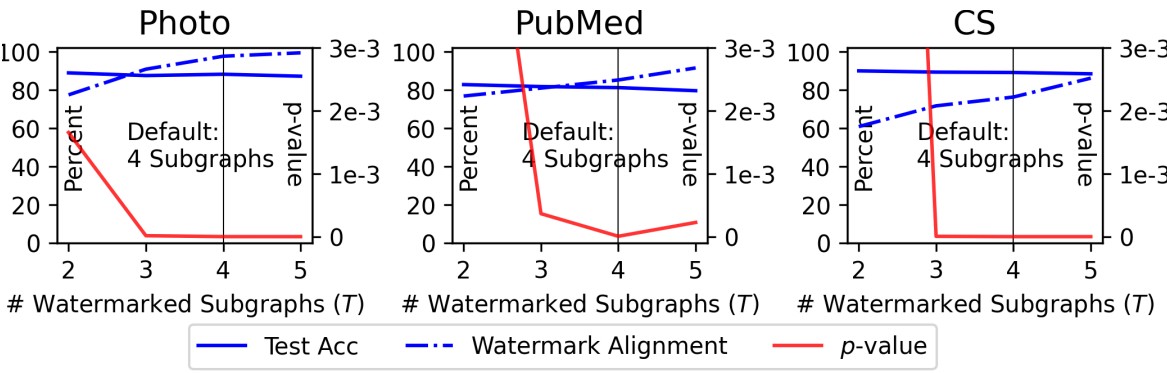

*Figure 4.* Watermarking metrics for varied $T$.

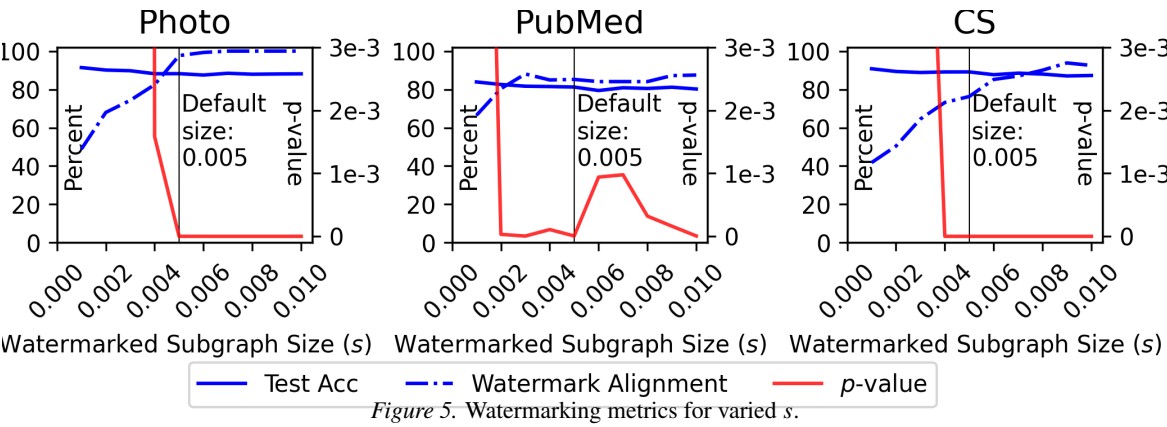

*Figure 5.* Watermarking metrics for varied $s$.

other attacks. Pruning compresses models by setting a portion of weights to zero (Li et al., 2016). Following (Liu et al., 2021a; Tekgul et al., 2021), we use *structured* pruning, targeting parameter tensor rows and columns based on $L_n$-norm importance scores (Paszke et al., 2019). Fine-tuning (Pan & Yang, 2010; Wang et al., 2021c) adapts already-trained models to a new task and may make GNNs "forget" watermarks, so it is commonly used for watermark robustness testing (Adi et al., 2018; Wang et al., 2020). We test our own method by continuing training on the *validation* dataset, $G^{val}$, at 0.1 times the original learning rate for 49 epochs. (See Appendix F for results with other learning rates and GNN architectures.)

Figure 2 shows pruning and fine-tuning results. Left: rates of 0.0 (no parameters pruned) to 1.0 (all pruned). In all datasets, the MI $p$-value only rises as classification accuracy drops, ensuring the owner detects pruning before it impacts the watermark. Right: classification accuracy and MI $p$-value during fine-tuning. CS has a near-zero MI $p$-value for 25 epochs; Photo, PubMed and Reddit2 have low MI $p$-values for the full duration, demonstrating robustness for extended periods of fine-tuning.

We assert that our method also resists data poisoning. Operating on randomly selected subgraphs without data-specific

assumptions (unlike backdoor methods), our watermark is embedded post-training-data selection, ensuring immunity to training data changes.

### 5.2.3. UNDETECTABILITY

**Brute-Force Search:** With Equations from Section 4.4, we demonstrate the infeasibility of a brute-force search for the watermarked subgraphs in our smallest dataset, Amazon Photo (4590 training nodes). Assume adversaries know the number ($T$) and size ($s$) of our watermarked subgraphs. With default $s = 0.005$, each subgraph has $ceil(0.005 \times 4590) = 23$ nodes – there are $\binom{4590}{23} = 6.1 \times 10^{61}$ possible subgraphs; with default $T = 4$, there are $\binom{\binom{4590}{23}}{4} = 5.8 \times 10^{245}$ possible subgraphs, making finding the *uniquely-convincing* set of watermarked subgraphs computationally infeasible.

**Random Search:** Figure 3 shows probabilities (Equation 13) that a randomly-chosen subgraph's nodes overlap with any watermarked subgraph, for varied sizes $s$. For $j = 1, \dots, 5$, or all $n_{sub}$ nodes, probability nears zero that a randomly-selected subgraph overlaps with a common watermarked subgraph by $\geq 3$ nodes (given our default watermark subgraph settings $T = 4$ and $s = 0.005$). (The exception is Reddit2, where $< 5$ nodes is an extremely small portion of the whole dataset.)

### 5.2.4. ABLATION STUDIES[6]

**Impact of the Number of Watermarked Subgraphs $T$:**
Figure 4 shows how $T$ affects watermark performance metrics. For all datasets, larger $T$ increases watermark alignment and a lower $p$-value, although test accuracy decreases slightly on Photo and PubMed. Notably, the default $T = 4$ is associated with a near-zero $p$-value in every scenario. Figure 10 in Appendix also shows the robustness results to removal attacks against varied $T$: we see that the watermarking method resists pruning attacks until test accuracy is affected, and fine-tuning attacks for at least 25 epochs.

**Impact of the Size of Watermarked Subgraphs $s$:** Figure 5 shows results with different sizes $s$. We see similar trends as Figure 4: watermarking is generally more effective, unique, and robust for larger $s$ values. There is again a slight trade-off between subgraph size and test accuracy. For $s \geq 0.003$, our method reaches near-zero $p$-values for all datasets, as well as increasing watermark alignment. Figure 11 in Appendix shows the robustness results: for all datasets, when $s > 0.005$, our method is robust against pruning attacks generally, and against fine-tuning attacks for at least 25 epochs.

**Impact of the Watermark Loss weight $r$:** Table 7 in Appendix shows the trade-off between the classification accuracy and the watermark effectiveness (suggested by p-value). Empirically, we can still achieve good classification performance while maintaining an effective watermark.

### 5.2.5. TIME COMPLEXITY

We also evaluate the impact of our watermarking on training time. The theoretical analysis can be found in C. Experiments on the four datasets show that including the watermark does not significantly increase the training duration. Detailed results are reported in Table 3.

## 6. Conclusion

We introduce the first GNN watermarking method using explanations, avoiding backdoor-based pitfalls with a statistically unambiguous watermark that resists data attacks. Demonstrating robustness against removal attempts and proving the statistical impossibility of locating watermarked subgraphs, our approach significantly advances GNN intellectual property protection.

---

[6]Note: Reddit2 is excluded from these ablation studies due to computational constraints, as the trends from the other three datasets are sufficient to demonstrate the effects of $s$ and $T$.

## Reproducibility Statement

Our datasets and implementation details are introduced in Appendix A and the codes are available at https://github.com/TIML-Group/graph_backdoor_explanation_detection.

## Impact Statement

This paper presents work whose goal is to advance the field of Machine Learning. There are many potential societal consequences of our work, none which we feel must be specifically highlighted here.

## Acknowledgments

This work was supported in part by the National Science Foundation under grants IIS-2246157, FMitF-2319243, ECCS-2216926, CCF-2331302, CNS-2241713, CNS-2339686, and the Department of Energy under grant DE-CR0000042. The project was also supported by computational resources provided by the NSF ACCESS and Argonne National Lab.

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

# Appendix

## A. Experimental Setup Details

**Hardware and Software Specifications.** All experiments were conducted on a MacBook Pro (Model Identifier: MacBookPro18,3; Model Number: MKGR3LL/A) with an Apple M1 Pro chip (8 cores: 6 performance, 2 efficiency) and 16 GB of memory, on macOS Sonoma Version 14.5. Models were implemented in Python with the PyTorch framework.

**Dataset Details.** Amazon Photo (simply "Photo" in this paper) is a subset of the Amazon co-purchase network (McAuley et al., 2015). Nodes are products, edges connect items often purchased together, node features are bag-of-words product reviews, and class labels are product categories. Photo has 7,650 nodes, 238,163 edges, 745 node features, and 8 classes. The CoAuthor CS dataset ("CS" in this paper) (Shchur et al., 2019) is a graph whose nodes are authors, edges are coauthorship, node features are keywords, and class labels are the most active fields of study by those authors. CS has 18,333 nodes, 163,788 edges, 6,805 node features, and 15 classes. Lastly, PubMed (Yang et al., 2016) is a citation network whose nodes are documents, edges are citation links, node features are TF-IDF weighted word vectors based on the abstracts of the papers, and class labels are research fields. The graph has 19,717 nodes, 88,648 edges, 500 features, and 3 classes. Reddit2 (Zeng et al., 2019) is a large social network where nodes represent posts, edges connect posts if the same user commented on both, node features are post embeddings, and class labels are communities. It contains 232,965 nodes, 114,848,857 edges, 602 features, and 41 classes. The details can be found in Table 2.

*Table 2.* Dataset Statistics

| Dataset | Nodes | Edges | Avg. Degree | Domain |
|---------|-------|-------|-------------|--------|
| CORA | 2,708 | 5,278 | 3.9 | Citation (small) |
| CiteSeer | 3,327 | 4,552 | 2.7 | Citation (small) |
| PubMed | 19,717 | 44,324 | 4.5 | Citation (medium) |
| Photo | 7,650 | 119,081 | 31.1 | Co-purchase |
| CS | 18,333 | 81,894 | 8.9 | Co-authorship |
| Reddit2 | 232,965 | 11,606,919 | 99.6 | Social (large) |

**Hyperparameter Setting Details.**

Classification training hyperparameters:

- Learning rate: 0.001-0.001

- Number of layers: 3

- Hidden Dimensions: 256-512

- Epochs: 100-300

Watermarking hyperparameters:

- Target significance level, $\alpha_{tgt}$: set to 1e-5 to ensure a watermark size that is sufficiently large.

- Verification significance level, $\alpha_v$: set to 0.01 to limit false verifications to under 1% likelihood.

- Watermark loss coefficient, $r$: set to values between 20-100, depending on the amount required to bring $L^{wmk}$ to a similar scale as $L^{clf}$ to ensure balanced learning.

- Watermark loss parameter $\epsilon$: set to values ranging from 0.01 to 0.1. Smaller values ensure that no watermarked node feature index has undue influence on watermark loss.

---

**Algorithm 1** Watermark Embedding

---

**Require:** Graph $G$, training nodes $\mathcal{V}^{tr}$, learning rate $\eta$, number of watermarked subgraphs $T$, watermark subgraph size $s$, hyperparameter $r$, target significance $\alpha_{tgt}$, watermark loss bound $\epsilon$.
**Ensure:** Trained and watermarked model $f$.
1: Initialize model $f$ and optimizer.
2: Compute $M$ using Eq. (11) with inputs $\alpha_{tgt}$, $T$, and feature dimension $F$.
3: Initialize watermark vector $\mathbf{w} \in \{-1, +1\}^M$ uniformly at random.
4: Let $n_{sub} = \lceil s \cdot |\mathcal{V}^{tr}| \rceil$.
5: Randomly sample $T$ subsets of $n_{sub}$ nodes from $\mathcal{V}^{tr}$ to form $G^{wmk}$.
6: Define remaining nodes as classification set $\mathcal{V}^{clf}$.
7: **for** epoch = 1 to #Epoch **do**
8: $\quad L^{clf} \leftarrow \mathcal{L}_{CE}(\mathbf{y}^{clf}, f_\Theta(\mathcal{V}^{clf}))$
9: $\quad L^{wmk} \leftarrow 0$
10: $\quad$ **for** $i = 1$ to $T$ **do**
11: $\quad\quad \mathbf{P}_i^{wmk} \leftarrow f_\Theta(\mathcal{V}_i^{wmk})$
12: $\quad\quad \mathbf{e}_i^{wmk} \leftarrow explain(\mathbf{X}_i^{wmk}, \mathbf{P}_i^{wmk})$
13: $\quad\quad L^{wmk} \leftarrow L^{wmk} + \sum_{j=1}^M \max(0, \epsilon - \mathbf{w}[j] \cdot \mathbf{e}_i^{wmk}[\mathbf{idx}[j]])$
14: $\quad$ **end for**
15: $\quad L \leftarrow L^{clf} + r \cdot L^{wmk}$
16: $\quad \Theta \leftarrow \Theta - \eta \nabla_\Theta L$
17: **end for**

---

## B. Gaussian Kernel Matrices

Define $\bar{\boldsymbol{K}}$ as a collection of matrices $\{\bar{\boldsymbol{K}}^{(1)}, \ldots, \bar{\boldsymbol{K}}^{(F)}\}$, where $\bar{\boldsymbol{K}}^{(k)}$ (size $N \times N$) is the centered and normalized version of Gaussian kernel matrix $\boldsymbol{K}^{(k)}$, and each element $\boldsymbol{K}_{uv}^{(k)}$ is the output of the Gaussian kernel function on the $k^{th}$ node feature for nodes $u$ and $v$:

$$\bar{\boldsymbol{K}}^{(k)} = \boldsymbol{H}\boldsymbol{K}^{(k)}\boldsymbol{H}/\|\boldsymbol{H}\boldsymbol{K}^{(k)}\boldsymbol{H}\|_F, \ \ \boldsymbol{H} = \boldsymbol{I}_N - \frac{1}{N}\mathbf{1}_N\mathbf{1}_N^T, \ \ \boldsymbol{K}_{uv}^{(k)} = \exp\left(-\frac{1}{2\sigma_x^2}\left(\mathbf{x}_u^{(k)} - \mathbf{x}_v^{(k)}\right)^2\right). \tag{15}$$

$\|\cdot\|_F$ is the Frobenius norm, $\boldsymbol{H}$ is a centering matrix (where $\boldsymbol{I}_N$ is an $N \times N$ identity matrix and $\mathbf{1}_N$ is an all-one vector of length $N$), and $\sigma_x$ is Gaussian kernel width. Now take the nodes' softmax scores $\mathbf{P} = [\mathbf{p}_1, \cdots, \mathbf{p}_N]$, and their Guassian kernel width, $\sigma_\mathbf{p}$. Define $\bar{L}$ as a centered and normalized $N \times N$ Gaussian kernel $L$, where $L_{uv}$ is the similarity between nodes $u$ and $v$'s softmax outputs:

$$\bar{L} = \boldsymbol{H}\boldsymbol{L}\boldsymbol{H}/\|\boldsymbol{H}\boldsymbol{L}\boldsymbol{H}\|_F, \quad L_{uv} = \exp\left(-\frac{1}{2\sigma_\mathbf{p}^2}\|\mathbf{p}_u - \mathbf{p}_v\|_2^2\right). \tag{16}$$

Let $\tilde{\boldsymbol{K}}$ be the $N^2 \times F$ matrix $[\text{vec}(\bar{\boldsymbol{K}}^{(1)}), \ldots, \text{vec}(\bar{\boldsymbol{K}}^{(F)})]$, where $\text{vec}(\cdot)$ converts each $N \times N$ matrix $\bar{\boldsymbol{K}}^{(k)}$ into a $N^2$-dimensional column vector. Similarly, we denote $\tilde{L} = \text{vec}(\bar{L})$ as the $N^2$-dimensional, vector form of the matrix $\bar{L}$. Also take $F \times F$ identity matrix $\boldsymbol{I}_F$ and regularization hyperparameter $\lambda$.

## C. Time Complexity Analysis

The training process involves optimizing for node classification and embedding the watermark. To obtain total complexity, we therefore need to consider two processes: forward passes with the GNN, and explaining the watermarked subgraphs.

**GNN Forward Pass Complexity.** The complexity of standard node classification in GNNs comes from two main processes: message passing across edges ($O(EF)$, where $E$ is number of edges and $F$ is number of node features), and weight multiplication for feature transformation ($O(NF^2)$, where $N$ is number of nodes). For $L$ layers, the time complexity of a forward pass is therefore:

$$O(L(EF + NF^2))$$

**Explanation Complexity.** Consider the Formula 3 for computing the explanation: $\mathbf{e} = explain(\mathbf{X}, \mathbf{P}) = (\tilde{K}^T \tilde{K} + \lambda I_F)^{-1} \tilde{K}^T \tilde{L}$. Remember that $\tilde{K}$ is an $N^2 \times F$ matrix, $I_F$ is a $F \times F$ matrix, and $\tilde{L}$ is a $N^2 \times 1$ vector. To compute the complexity of this computation, we need the complexity of each subsequent order of operations:

1. Multiplying $\tilde{K}^T \tilde{K}$ (an $O(F^2 N^2)$ operation, resulting in an $F \times F$ matrix)

2. Obtaining and adding $\lambda I_F$ (an $O(F^2)$ operation, resulting in an $F \times F$ matrix)

3. Inverting the result (an $O(F^3)$ operation, resulting in an $F \times F$ matrix)

4. Multiplying by $\tilde{K}^T$ (an $O(F^2 N^2)$ operation, resulting in an $F \times N^2$ matrix)

5. Multiplying the result by $\tilde{L}$ (an $O(F^2 N^2)$ operation, resulting in an $N^2 \times 1$ vector)

The total complexity of a single explanation is therefore $O(F^2 N^2) + O(F^2) + O(F^3) + O(F^2 N^2) + O(F^2 N^2) = O(F^2 N^2 + F^3)$. For obtaining explanations of $T$ subgraphs during a given epoch of watermark embedding, the complexity is therefore:

$$O(T(F^2 N^2 + F^3))$$

**Total Complexity.** The total time complexity over $i$ epochs is therefore:

$$O \left( i \times \left( L(EF + NF^2) + T(F^2 N^2 + F^3) \right) \right)$$

The above calculation is based on general graph notations in Section 3.1. The actual explanation complexity is much smaller. For the ridge regression, as mentioned in Section 4.1, we use only the nodes from the subgraphs $\{G_1^{wmk}, \ldots, G_T^{wmk}\}$. Here, $N$ is the number of nodes in $\{G_1^{wmk}, \ldots, G_T^{wmk}\}$. Additionally, as mentioned in Section 4.3, the original feature space of size $F$ is masked to a subset of size $M$, which reduces the complexity of backpropagating the watermark loss.

**Training Duration (Wall Time).** We evaluate the training duration on the Photo, PubMed and CS datasets under both watermarked and non-watermarked settings. The corresponding training times are reported in Table 3. In both cases, the models are trained using 7 subgraphs. The architecture comprises three layers with a hidden dimension of 256. The results suggest that introducing the watermark does not increase the training time to a prohibitive degree.

*Table 3.* Model Training time (in seconds) with and without watermarking.

| Dataset | Architecture | Epochs | Without Watermark (s) | With Watermark (s) |
|---------|--------------|--------|-----------------------|--------------------|
| Photo | GCN | 300 | 91.72 | 147.25 |
| Photo | SAGE | 300 | 109.06 | 167.28 |
| PubMed | GCN | 200 | 59.20 | 96.84 |
| PubMed | SAGE | 200 | 65.18 | 98.98 |
| CS | GCN | 90 | 380.45 | 723.75 |
| CS | SAGE | 90 | 387.98 | 911.32 |
| Reddit2 | GCN | 90 | 7651.30 | 13729.49w |
| Reddit2 | SAGE | 90 | 6557.06 | 15198.00 |

## D. Normality of Matching Indices Distribution

Our results rely on the $z$-test to demonstrate the significance of the $MI$ metric. To confirm that this test is appropriate, we need to demonstrate that the $MI$ values follow a normal distribution. Table 4 shows the results of applying the Shapiro-Wilk (Ghasemi & Zahediasl, 2012) normality test to $MI$ distributions obtained under different GNN architectures and datasets. The results show $p$-values significantly above 0.1, indicating we cannot reject the null hypothesis of normality.

*Table 4.* Shapiro-Wilk Test p-values

| Dataset | SAGE | SGC | GCN |
|---------|------|-----|-----|
| Photo   | 0.324 | 0.256 | 0.345 |
| CS      | 0.249 | 0.240 | 0.205 |
| PubMed  | 0.249 | 0.227 | 0.265 |

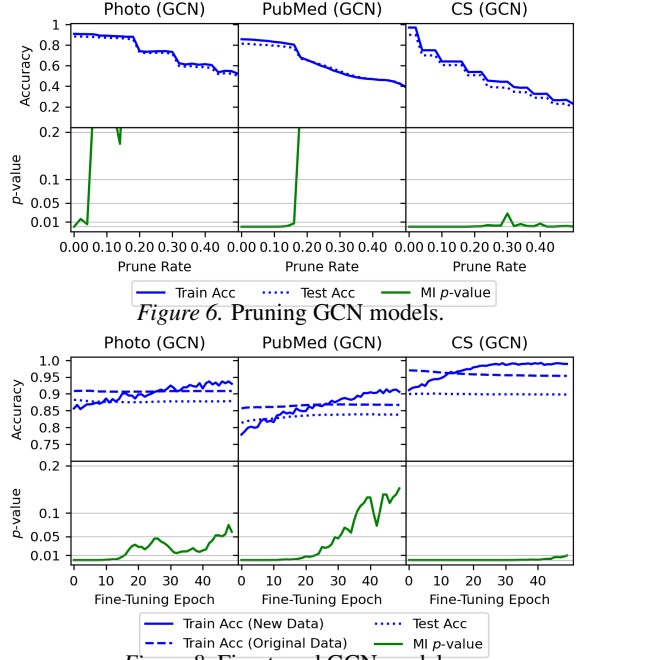

*Figure 6.* Pruning GCN models.

*Figure 8.* Fine-tuned GCN models.

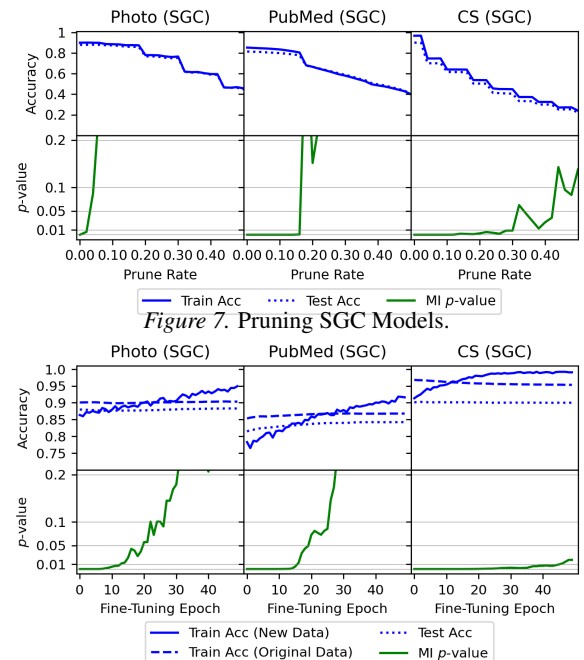

*Figure 7.* Pruning SGC Models.

*Figure 9.* Fine-tuned SGC models.

# E. Robustness of Our Method Against Different Types of Attacks

**Fine-tuning and pruning under more GNN architectures.** The main paper mainly show results on GraphSAGE (Hamilton et al., 2018). Here, we also explore GCN (Kipf & Welling, 2017) and SGC (Wu et al., 2019). Figure 6-Figure 9 shows the impact of fine-tuning and pruning attacks results on our watermarking method under these two architectures. Watermarked GCN and SGC models fared well against fine-tuning attacks for the Photo and CS datasets, but less so for PubMed; meanwhile, these models were robust against pruning attacks for Pubmed and CS datasets, but not Photo. Since the owner can assess performance against these removal attacks prior to deploying their model, they can simply a matter of training each type as effectively as possible and choosing the best option. In our case, GraphSAGE fared best for our three datasets, but GCN and SGC were viable solutions in some cases.

**Fine-Tuning and Pruning under varied watermark sizes.** Figures 10 and 11 show the robustness of our method to fine-tuning and pruning removal attacks when $T$ and $s$ are varied. We observe that, for $T \geq 4$ and $s \geq 0.005$ — our default values — pruning only affects MI $p$-value after classification accuracy has already been affected; at this point the pruning attack would be detected by model owners regardless. Similarly, across all datasets, for $T \geq 4$ and $s \geq 0.005$, our method demonstrates robustness against the fine-tuning attack for at least 25 epochs.

**Fine-Tuning under varied learning rates.** Our main fine-tuning results (see Figure 2) scale the learning rate to 0.1 times its original training value. Figure 12 additionally shows results for learning rates scaled to 1× and 10× the original training rates. The results for scaling the learning rate by 1× show that larger learning rates quickly remove the watermark. However, these figures also demonstrate that, by the time training accuracy on the fine-tuning dataset has reached an acceptable level of accuracy, the accuracy on the original training set drops significantly, which diminishes the usefulness of the fine-tuned model on the original task. For larger rates (10×), the watermark is removed almost immediately, but the learning trends and overall utility of the model are so unstable that the model is rendered useless. Given this new information, our default choice to fine-tune at 0.1× the original learning rate is the most reasonable scenario to consider.

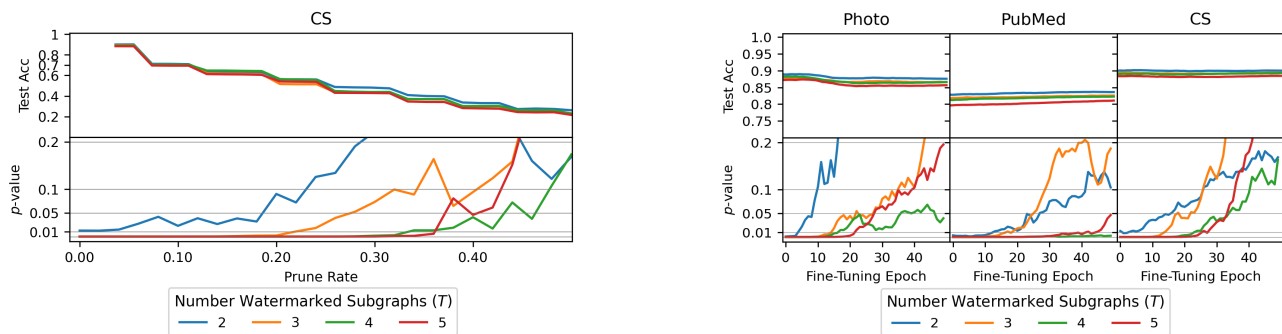

*Figure 10.* Pruning and fine-tuning attacks against varied number of watermarked subgraphs ($T$)

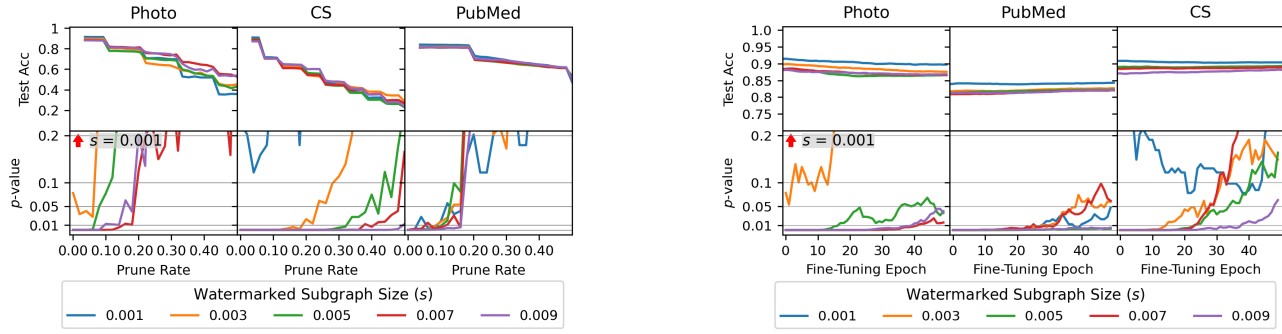

*Figure 11.* Pruning and fine-tuning attacks against varied sizes of watermarked subgraphs ($s$)

**Watermark effectiveness after model merge.** Model merge is a common strategy to merge two models that share the optimization trajectory. We conduct experiments that show the robustness of the watermark after the model merge. Following the idea in (Wortsman et al., 2022), we generate a new model by averaging parameters from the watermark model and the fine-tuned watermark model. Table 5 shows effective watermark verification (suggested by p-value) in the merged model.

**Knowledge Distillation Attack.** An important future direction is to safeguard our method against model extraction attacks (Shen et al., 2022), which threaten to steal a model's functionality without preserving the watermark. One form of model extraction attack is knowledge distillation attack (KD) (Gou et al., 2021).

Knowledge distillation has two models: the original "teacher" model, and an untrained "student" model. During each epoch, the student model is trained on two objectives: (1) correctly classify the provided input, and (2) mimic the teacher model by mapping inputs to the teacher's predictions. The student therefore learns to map inputs to the teacher's "soft label" outputs (probability distributions) alongside the original hard labels; this guided learning process leverages the richer information in the teacher's soft label outputs, which capture nuanced relationships between classes that hard labels cannot provide. By focusing on these relationships, the student model can generalize more efficiently and achieve comparable performance to the teacher with a smaller model and fewer parameters, thus reducing complexity.

We find that in the absence of a strategically-designed defense, the knowledge distillation attack successfully removes our watermark ($p > 0.05$). This is unsurprising, since model distillation maps inputs to outputs but ignores mechanisms that lead to auxiliary tasks like watermarking.

To counter this, we outline a defense mechanism that would incorporate watermark robustness to knowledge distillation.

*Table 5.* Accuracy of the Watermarked and Merged Models and the MI p-value of the Merged Model

| Architecture | **Watermark Model Acc.** (train/valid/test) | **Merged Model Acc.** (train/valid/test) | **p-value** |
|---|---|---|---|
| GCN | 0.862/0.807/0.801 | 0.861/0.816/0.813 | 9.633e-7 |
| SAGE | 0.905/0.790/0.776 | 0.897/0.801/0.786 | 0.007 |
| Transformer | 0.904/0.790/0.782 | 0.906/0.798/0.790 | 0.008 |

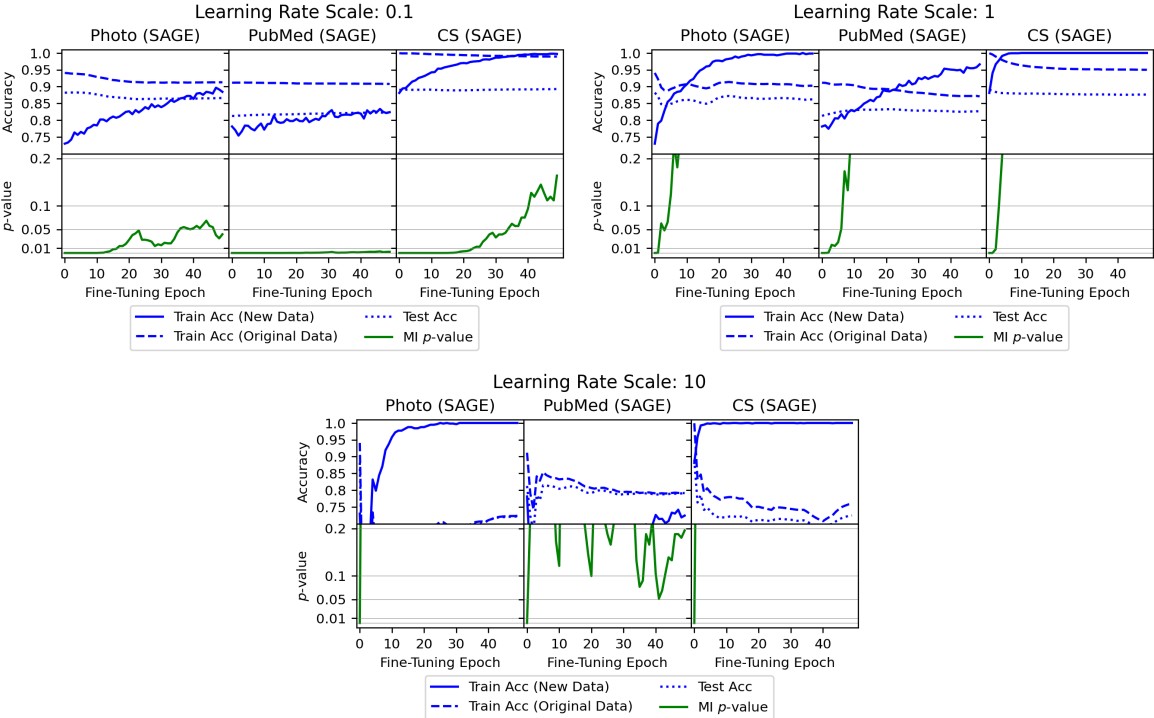

*Figure 12.* Fine-tuning results at increased learning rates (SAGE architecture).

This method strategically injects the watermark pattern directly into the teacher model's soft-label output space during the distillation phase, thereby enforcing its mandatory inheritance by the student model. The details of the mechanism is as follows:

1. **Watermark Vector Generation.** The mechanism begins by generating a highly structured perturbation vector ($\mathbf{v}$) for each input graph, where the dimensionality matches the number of classification classes ($C$). The perturbation vector $\mathbf{v}$ is constructed via a linear transformation involving two private components: (1) Gaussian projection matrix ($\mathbf{A}$): A randomly generated matrix $\mathbf{A} \in \mathbb{R}^{C \times M}$ sampled from a Gaussian Distribution. (2) The watermark ground truth ($\mathbf{w}$): The watermark pattern $\mathbf{w} \in \mathbb{R}^M$, where $M$ is the length of watermark. The perturbation vector is computed as:

$$\mathbf{v} = \mathbf{A}\mathbf{w}$$

2. **Logits Perturbation and Normalization.** The perturbation vector $\mathbf{v}$, whose dimension $C$ matches the logits space, is then normalized by using $L_2$ normalization to ensure its magnitude is minute. This normalization step is crucial for achieving utility preservation. The normalized vector $\mathbf{v}'$ is then added to the teacher model's original logits ($\mathbf{z}_T$) for each node:

$$\mathbf{z}'_T = \mathbf{z}_T + \mathbf{v}'$$

3. **Knowledge Distillation Enforcement.** All the steps above are performed before the KD process. During the KD process, the student model is forced to minimize the Kullback-Leibler (KL) divergence between its own soft-label predictions ($\mathbf{P}_S$) and the perturbed soft-label predictions ($\mathbf{P}'_T$) derived from $\mathbf{Z}'_T$.

$$\mathcal{L}_{KD} = T^2 \cdot \mathrm{KL}\left(\mathrm{LogSoftmax}\left(\frac{\mathbf{z}_S}{T}\right) \| \mathrm{Softmax}\left(\frac{\mathbf{z}'_T}{T}\right)\right)$$

Because the $\mathbf{v}'$ vector is numerically negligible after normalization, adding it to the logits causes no significant degradation (i.e., minimal utility loss) to the teacher's classification accuracy. However, $\mathbf{v}'$ contains the secret linear structure defined by $\mathbf{A}$ and $\mathbf{w}$.

The student model, in its attempt to precisely mimic the rich information in the perturbed soft labels ($\mathbf{P}'_T$), is consequently forced to inherit the linear relationship defined by the secret watermark pattern $\mathbf{w}$, thereby achieving Transferable Watermarking.

With this mechanism, we evaluate the effectiveness of our method against KD attacks. The results on the Photo dataset using the GCN architecture can be found in Table 6. It demonstrates that our watermark exhibits robust transferability to the student model. The student model maintained high classification accuracy on the task, confirming that the normalized logits perturbation ($\mathbf{v}'$) did not significantly compromise the model's primary utility. The measured Matching Index (MI) for the watermarked subgraphs yielded a low P-value ($P \leq 0.05$), achieving the required statistical significance for ownership verification. This confirms that the watermark pattern was successfully inherited and preserved within the student model's feature attribution space despite the model compression achieved through KD.

*Table 6.* Experimental Validation of Robust Watermark Transfer under Knowledge Distillation Attack.

| # Subgraph Collections | 6 | | 7 | |
|---|---|---|---|---|
| | without | **with** | without | **with** |
| **p-value** | 0.352 | **0.037** | 0.225 | **0.025** |
| **Acc (train/test)** | 0.904/0.883 | **0.904/0.880** | 0.904/0.881 | **0.903/0.880** |

## F. Additional Results

**Ablation Study of Hyperparameter $r$.** As mentioned in Section 5.2.4, we observe a trade-off between the classification accuracy and the watermark effectiveness. Here, we include the corresponding empirical results. $r$ represents the weight of the watermark loss.

*Table 7.* Effect of $r$ on accuracy and MI $p$-value across datasets.

| Dataset | r | Accuracy (Train/Valid/Test) | p-value |
|---|---|---|---|
| Photo | 0 | 0.957 / 0.937 / 0.948 | – |
| Photo | 1 | 0.955 / 0.937 / 0.950 | 0.23 |
| Photo | 20 | 0.951 / 0.937 / 0.946 | 0.012 |
| Photo | 100 | 0.937 / 0.934 / 0.939 | $1.00 \times 10^{-4}$ |
| Photo | 200 | 0.928 / 0.933 / 0.931 | 0.002 |
| PubMed | 0 | 0.899 / 0.876 / 0.871 | – |
| PubMed | 1 | 0.894 / 0.876 / 0.865 | $4.67 \times 10^{-6}$ |
| PubMed | 20 | 0.874 / 0.867 / 0.860 | 0 |
| PubMed | 100 | 0.813 / 0.811 / 0.806 | 0 |
| PubMed | 200 | 0.730 / 0.723 / 0.735 | 0 |

**More Results on Effectiveness and Uniqueness.** Table 1 in the main paper shows the test accuracy, watermark alignment, and MI $p$-values of our experiments with the default value of $T = 4$. In Table 8, we additionally present the results for $T = 2, T = 3$, and $T = 5$. The results show MI $p$-values below 0.001 across all configurations when $T \geq 3$. They also show increasing watermark alignment with increasing $T$, however, with a slight trade-off in classification accuracy: when increasing from $T = 2$ to $T = 5$, watermark alignment increases, but train and test classification accuracy decreases by an average of 1.44% and 2.13%, respectively; despite this, both train and test classification accuracy are generally high across all datasets and models.

**Comparison with Backdoor-based Watermark Methods.** To empirically evaluate our approach compared to backdoor-based GNN watermarking, we compare our method with the representative work by Xu et al. (Xu et al., 2023) across three key dimensions: verification significance, robustness against fine-tuning, and targeted fine-tuning based on backdoor example detection.

We re-implemented the work (Xu et al., 2023) from scratch based on their paper, as no official source code is available. Our implementation faithfully follows their key design choices and optimized hyperparameter (0.15 poisoning rate for

*Table 8.* Watermarking results for varied $T$. Each value averages 5 trials with distinct random seeds.

| | | Number of Subgraphs ($T$) | | | | | | | | | | |
|---|---|---|---|---|---|---|---|---|---|---|---|---|
| | | 2 | | | 3 | | | 4 | | | 5 | | |
| Dataset | GNN | Acc (Trn/Tst) | Wmk Align | MI $p$-val | Acc (Trn/Tst) | Wmk Align | MI $p$-val | Acc (Trn/Tst) | Wmk Align | MI $p$-val | Acc (Trn/Tst) | Wmk Align | MI $p$-val |
| **Photo** | GCN | 92.5/89.7 | 73.0 | 0.087 | 91.5/88.9 | 86.1 | <0.001 | 90.9/88.3 | 91.4 | <0.001 | 90.6/88.2 | 95.2 | <0.001 |
| | SGC | 92.0/89.4 | 73.8 | 0.111 | 91.0/88.7 | 82.5 | <0.001 | 90.1/88.0 | 91.8 | <0.001 | 89.7/87.4 | 99.4 | <0.001 |
| | SAGE | 95.4/88.9 | 77.4 | 0.002 | 94.4/87.5 | 90.9 | <0.001 | 94.1/88.2 | 97.7 | <0.001 | 93.9/87.2 | 99.4 | <0.001 |
| **PubMed** | GCN | 87.0/83.7 | 75.4 | 0.003 | 85.9/82.1 | 86.6 | <0.001 | 85.7/81.4 | 91.5 | <0.001 | 85.6/81.4 | 90.2 | <0.001 |
| | SGC | 86.7/83.1 | 79.7 | <0.001 | 85.8/81.6 | 83.8 | <0.001 | 85.3/81.4 | 88.9 | <0.001 | 84.6/80.0 | 92.9 | <0.001 |
| | SAGE | 91.9/82.8 | 76.8 | 0.009 | 91.3/81.8 | 81.0 | <0.001 | 91.1/81.2 | 85.2 | <0.001 | 90.1/79.6 | 91.5 | <0.001 |
| **CS** | GCN | 97.1/90.3 | 56.8 | 0.562 | 96.8/89.9 | 67.5 | <0.001 | 96.8/89.8 | 73.8 | <0.001 | 96.9/90.0 | 78.9 | <0.001 |
| | SGC | 97.2/90.3 | 57.1 | 0.003 | 96.8/89.9 | 67.7 | <0.001 | 96.7/90.1 | 74.5 | <0.001 | 96.6/89.8 | 77.8 | <0.001 |
| | SAGE | 99.9/90.2 | 61.5 | 0.233 | 99.9/89.4 | 73.3 | <0.001 | 99.9/88.9 | 78.2 | <0.001 | 99.9/88.3 | 84.0 | <0.001 |

*Table 9.* Verification Significance: Our Method vs (Xu et al., 2023)

| Dataset | Method | Test acc | Verification (5 seeds) |
|---|---|---|---|
| CORA | Ours | $0.849 \pm 0.005$ | **5/5 pass**, $z = 13.16 \pm 1.51$ |
| CORA | Xu et al | $0.835 \pm 0.025$ | 4/5 pass, backdoor acc = $0.866 \pm 0.265$[†] |
| CiteSeer | Ours | $0.731 \pm 0.009$ | **5/5 pass**, $z = 12.43 \pm 0.75$ |
| CiteSeer | Xu et al | $0.729 \pm 0.006$ | 5/5 pass, backdoor acc = $1.000 \pm 0.000$ |
| PubMed | Ours | $0.824 \pm 0.002$ | **5/5 pass**, $z = 7.74 \pm 1.10$ |
| PubMed | Xu et al | $0.851 \pm 0.016$ | 4/5 pass, backdoor acc = $0.901 \pm 0.198$[†] |

†: Seed 3 failed because the randomized feature value applied to the trigger examples was too small to be distinguished from other regular instances.

GraphSAGE) with a single scalar trigger value drawn uniformly from [0,1] and a fixed target label for all poisoned nodes. We utilize the two datasets used in (Xu et al., 2023) (CORA, CiteSeer) and one of the datasets we utilized in our work (PubMed).

Our framework shows similar classification test accuracy as (Xu et al., 2023). In terms of verification significance, (Xu et al., 2023) failed at seed 3 because the randomized feature value applied to the trigger examples was too small to be distinguished from other regular instances. Our framework shows consistent significance in the verification process in Table 9.

As shown in Table 10, both methods survive fine-tuning across all three datasets. Beyond the numbers, we highlight a fundamental security distinction. Backdoor watermarking methods embed the mark as an explicit trigger. The trigger is a concrete artifact present in the training data and is detectable via feature outlier analysis or activation clustering. Our method never modifies the training data, eliminating this risk entirely.

To illustrate this, inspired by (Chen et al., 2018a), we show that (Xu et al., 2023)'s trigger is detectable in raw feature space and can be surgically removed. We run K-Means (k=2) per class on PCA-reduced training features and select the most anomalously separated class via top-1 silhouette score. Within that class, the poison cluster is identified by a const-nonzero heuristic (features with near-zero variance and non-zero mean), directly recovering the trigger positions. Detected nodes are relabeled using model predictions on trigger-zeroed features, then the model is fine-tuned on the corrected data. This forces re-associating the trigger with the correct class. Results across all three datasets (seeds 1–5) can be found in Table 11.

**The Stability of the Explanation Mechanism.** We conduct two experiments that demonstrate the stability and robustness of our method against variation in initialization during training as well as perturbation on node features. First, we retrain with seeds 1-5 on data sampling and initialization. Secondly, during inference, we inject perturbations on each node feature by independent Gaussian noise with empirical std as 5% / 10% / 20% of the feature values' std. In both cases, we observed a small variation in test accuracy and verification p-value. The results can be found in Tables 12- 13.

**Comparison with other Explainer-based Method.** (Shao et al., 2024) enables watermarking for image data using explainer-produced artifacts, it cannot be directly adapted to GNNs due to two fundamental limitations: it fails to utilize topological information and lacks a mechanism to aggregate results across different categories. In an empirical comparison, we achieve comparable classification accuracy while producing a statistically significant watermark, whereas (Shao et al.,

*Table 10.* Our Method vs (Xu et al., 2023): Robustness against Fine-Tuning.

| Dataset | Method | Test acc (post-FT) | Watermark (post-FT) |
|---|---|---|---|
| CORA | Ours | $0.856 \pm 0.013$ | $z = 7.83 \pm 1.42$, $p = 1.6 \times 10^{-10}$ |
| CORA | Xu et al | $0.845 \pm 0.022$ | backdoor acc = $0.847 \pm 0.293^{\dagger}$ |
| CiteSeer | Ours | $0.722 \pm 0.010$ | $z = 3.56 \pm 0.79$, $p = 1.1 \times 10^{-3}$ |
| CiteSeer | Xu et al | $0.736 \pm 0.008$ | backdoor acc = $1.000 \pm 0.000$ |
| PubMed | Ours | $0.829 \pm 0.008$ | $z = 5.27 \pm 1.86$, $p = 9.8 \times 10^{-4}$ |
| PubMed | Xu et al | $0.830 \pm 0.037$ | backdoor acc = $0.823 \pm 0.353^{\dagger}$ |

$\dagger$: Seed 3 failed because the randomized feature value applied to the trigger examples was too small to be distinguished from other regular instances.

*Table 11.* Targeted Fine-tuning Attack on (Xu et al., 2023).

| Dataset | Detection | Test acc after | Backdoor Acc before | Backdoor Acc after |
|---|---|---|---|---|
| CORA | 4/5 ✓, 1/5 weak$^{\dagger}$ | $0.837 \pm 0.031$ | $0.866 \pm 0.265$ | $\mathbf{0.224 \pm 0.064}$ |
| CiteSeer | 5/5 ✓ | $0.744 \pm 0.014$ | $1.000 \pm 0.000$ | $\mathbf{0.142 \pm 0.018}$ |
| PubMed | 4/5 ✓, 1/5 ×$^{\ddagger}$ | $0.848 \pm 0.014$ | $0.901 \pm 0.198$ | $\mathbf{0.236 \pm 0.136}$ |

$^{\dagger}$ CORA seed 3: trigger value $\approx 0.024$, backdoor already broken before attack (0.337).
$^{\ddagger}$ PubMed seed 3: trigger value $\approx 0.007$, no constant non-zero cluster found; thus backdoor acc before = after = 0.506.

*Table 12.* Stability Under Different Initializations (PubMed, GCN, Seed 1-5).

| Metric | Mean | Std |
|---|---|---|
| Test Acc | 0.8024 | 0.0094 |
| Watermark P-value | $2.14 \times 10^{-6}$ | $2.30 \times 10^{-6}$ |

*Table 13.* Stability Under Input Perturbation (PubMed, GCN, Seeds 1–5).

| Perturbation $\alpha$ | Watermark P-value (mean) | Watermark P-value (std) |
|---|---|---|
| 0.0 (baseline) | $2.14 \times 10^{-6}$ | $2.30 \times 10^{-6}$ |
| 0.05 | $8.22 \times 10^{-6}$ | $1.50 \times 10^{-5}$ |
| 0.10 | $8.63 \times 10^{-6}$ | $9.43 \times 10^{-6}$ |
| 0.20 | $1.08 \times 10^{-4}$ | $2.06 \times 10^{-4}$ |

*Table 14.* Our Method vs. Shao et al. (PubMed, GCN, seed=1).

| Method | Test Acc | Watermark P-value |
|---|---|---|
| Our method | 0.795 | $4.00 \times 10^{-10}$ |
| Shao et al. | 0.806 | 0.8879 |

2024) fails the watermark significance test entirely, as shown in Table 14.

# G. Future Directions

While we have primarily provided results for the node-classification case, we believe much of our logic can be extended to other graph learning tasks, including edge classification and graph classification. Our method embeds the watermark into explanations of predictions on various graph features. Specifically, for node predictions, we obtain feature attribution vectors for the $n \times F$ node feature matrices of $T$ target subgraphs, with a loss function that penalizes deviations from the watermark. This process can be adapted to link prediction and graph classification tasks as long as we can derive $T$ separate

$n \times F$ feature matrices, where $n$ represents the number of samples per group and $F$ corresponds to the number of features for the given data structure (e.g., node, edge, or graph). Below, we outline how this extension applies to different classification tasks:

1. **Node Classification:** The dataset is a single graph. Subgraphs are formed by randomly selecting $n = s \cdot |\mathcal{V}^{tr}|$ nodes from the training set (where $|\mathcal{V}^{tr}|$ is the number of training nodes and $s$ is a proportion of that size). (Note: in this case, $n$ is equal to the value $n_{sub}$ referenced previously in the paper.) For each subgraph:

   - The $n \times F$ node feature matrix represents the input features ($F$ is the number of node features).
   - The $n \times 1$ prediction vector contains one label per node.
   - These inputs are used in a ridge regression problem to produce a feature attribution vector for the subgraph.
   - With $T$ subgraphs, we generate $T$ explanations.

2. **Link Prediction:** Again, the dataset is a single graph. Subgraphs are formed by randomly selecting $n = s \cdot |\mathcal{E}^{tr}|$ edges. For each subgraph:

   - Each row in the $n \times F$ feature matrix represents the features of a single link. These features are derived by combining the feature vectors of the two nodes defining the link, using methods such as concatenation or averaging. The resulting feature vector for each link has a length of $F$.
   - The $n \times 1$ prediction vector contains one label per edge.
   - These inputs are used in a ridge regression problem to produce a feature attribution vector for the subgraph.
   - As with node classification, we generate $T$ explanations for $T$ subgraphs.

3. **Graph Classification:** For graph-level predictions, the dataset $\mathcal{D}^{tr}$ is a collection of graphs. We extend the above pattern to $T$ collections of $n = s \cdot |\mathcal{D}^{tr}|$ subgraphs, where each subgraph is drawn from a different graph in the training set. Specifically:

   - Each subgraph in a collection is summarized by a feature vector of length $F$ (e.g., by averaging its node or edge features).
   - For a collection of $n$ subgraphs, we construct:
     - An $n \times F$ subgraph feature matrix, where each row corresponds to a subgraph in the collection.
     - An $n \times 1$ prediction vector, containing one prediction per subgraph.
   - These inputs are used in a ridge regression problem to produce a feature attribution vector for the collection.
   - With $T$ collections of $n$ subgraphs, we produce $T$ explanations.

By consistently framing each task as $T$ groups of $n \times F$ data points, our method provides a unified approach while adapting $F$ to the specific task requirements.

For instance, Table 15 provides sample results on graph classification using the MUTAG dataset; the results demonstrate that our method is effective beyond node classification.

*Table 15.* Watermarking results: graph classification

| # Subgraph Collections | 4 | 5 | 6 |
|---|---|---|---|
| p-value | 0.039 | 0.037 | <0.001 |
| Acc (train/test) | 0.915/0.900 | 0.954/0.929 | 0.915/0.893 |

## H. The Use of Large Language Models

In this work, large language models (LLMs) were not used in any part of the methodology, data analysis, or experiments. Their role was solely limited to polishing the language and improving the readability of the manuscript. All scientific ideas, experimental designs, and results are entirely the work of the authors.

---

**Algorithm 2** Ownership Verification

---

**Require:** A trained GNN $f$ from Alg. 1, graph $G$ with training nodes $\mathcal{V}^{tr}$, $T$ candidate subgraphs of size $n_{sub}$, verification significance level $\alpha_v$, and $I$ simulations.

**Ensure:** Ownership verification verdict.

1: **Phase I: Obtain distribution of naturally-occurring matches**
2: Define random subgraphs $\mathcal{S} = \{G_1^{rand}, \ldots, G_D^{rand}\}$, where each subgraph has $n_{sub} = \lceil s \cdot |\mathcal{V}^{tr}| \rceil$ nodes sampled uniformly at random from $\mathcal{V}^{tr}$. Choose $D$ sufficiently large (e.g., $D > 100$).
3: For each $1 \leq i \leq D$, compute binarized explanations $\widehat{\mathbf{E}}_i^{rand}$ using Eq. (6).
4: Initialize empty list $matchCounts \leftarrow \emptyset$.
5: **for** $i = 1$ to $I$ **do**
6:     Randomly sample $T$ distinct indices $\{idx_1, \ldots, idx_T\}$ from $\{1, \ldots, D\}$.
7:     For each sampled $idx_k$, obtain node set $\mathcal{V}_{idx_k}^{rand}$ and features $\mathbf{X}_{idx_k}^{rand}$.
8:     Compute $\widehat{\mathbf{E}}_{idx_k}^{rand} = \text{sign}\left(explain(\mathbf{X}_{idx_k}^{rand}, f(\mathcal{V}_{idx_k}^{rand}))\right)$ for all $k$.
9:     Compute match index MI over $\{\widehat{\mathbf{E}}_{idx_1}^{rand}, \ldots, \widehat{\mathbf{E}}_{idx_T}^{rand}\}$ using Eq. (7), and append to $matchCounts$.
10: **end for**
11: Compute mean and standard deviation:

$$\mu_{nat} = \frac{1}{I} \sum_{i=1}^{I} matchCounts[i], \qquad \sigma_{nat} = \sqrt{\frac{1}{I} \sum_{i=1}^{I} (matchCounts[i] - \mu_{nat})^2}.$$

12: **Phase II: Significance Testing**
13: Consider null hypothesis $H_0$: the observed MI across candidate subgraphs $\{\widehat{\mathbf{E}}_i^{cdt}\}_{i=1}^{T}$ comes from the naturally-occurring population.
14: **for** $i = 1$ to $T$ **do**
15:     Compute predictions $\mathbf{P}_i^{cdt} = f(\mathcal{V}_i^{cdt})$ and obtain features $\mathbf{X}_i^{cdt}$.
16:     Compute binarized explanation

$$\widehat{\mathbf{E}}_i^{cdt} = \text{sign}\left(explain(\mathbf{X}_i^{cdt}, \mathbf{P}_i^{cdt})\right).$$

17: **end for**
18: Compute $\text{MI}^{cdt}$ over $\{\widehat{\mathbf{E}}_i^{cdt}\}_{i=1}^{T}$ using Eq. (14).
19: Compute $z$-test statistic and p-value:

$$z_{test} = \frac{\text{MI}^{cdt} - \mu_{nat}}{\sigma_{nat}}, \qquad p = 1 - \Phi(z_{test}).$$

20: **if** $p < \alpha_v$ **then**
21:     Reject $H_0$ and verify ownership.
22: **else**
23:     Fail to reject $H_0$; insufficient ownership evidence.
24: **end if**

---

