# OpenReview forum: "Watermarking Graph Neural Networks via Explanations for Ownership Protection"
_ICML.cc/2026/Conference — ICML 2026 regular_

### Official Review · Reviewer_Qe8R · 2026-03-05

**Soundness:** 3
**Presentation:** 3
**Significance:** 4
**Originality:** 3
**Overall Recommendation:** 5
**Confidence:** 3

**Summary:**

This paper presents the first explanation-based watermarking technique for GNNs that is robust to fine-tuning and pruning attacks. Their main insight is that aligned watermarked subgraph explanations with watermarked patterns must yield statistically-significant similarity between explanations (explanation of a GNN’s node predictions is the feature attribution vector e) that is unlikely to occur in alternate solutions. Empirically, their method is resistant to removal attacks, including model pruning, fine-tuning, model merging, and knowledge distillation. They also show that their method is resistant to data poisoning attacks.

**Compliance With Llm Reviewing Policy:**

Affirmed.

**Final Justification:**

My final recommendation to Accept is based on the paper’s significant contribution as the first explanation-based watermarking framework for GNNs, supplying a robust solution for ownership verification in real-world, black-box settings. The work demonstrates high originality and significance by successfully defending against a variety of removal and data poisoning attacks without compromising the model's primary classification performance. While I initially had concerns regarding the potential for ownership ambiguity if node features shared similar importance levels, the authors’ rebuttal effectively addressed this by clarifying that verification is not dependent on absolute alignment, but on joint statistical significance across multiple subgraphs relative to an empirical null distribution. This explanation supported my confidence in the soundness of the methodology. Given the technical solidity of the approach and its clear potential for impact across multiple sub-areas, I believe the paper fulfills the high standards for publication.

**Key Questions For Authors:**

- If there are node features with similar importance, is it possible to have ambiguous explanations that can be used to claim ownership?
- How to prevent the case when the same or similar node features lead to similar statistical significance, since, as I understood, the importance of each input feature for the GNN’s predictions? Since you mentioned in the footnotes that alignment must simply be “good enough.”

**Limitations:**

- From Figure 3, we notice that as the subgraph size increases, the probability that a randomly-chosen subgraph overlaps with a watermarked subgraph increases, but inversely with the number of nodes that overlap. So, is there a minimum size of subgraph explanation that should be enforced? Since making big enough subgraph explanations can lead to similar issues (potential limitation), can high statistical significance be a problem?

**Strengths And Weaknesses:**

Strength:
+ first watermarking method for GNNs
+ black-box assumption while ownership verification showing real-world deployment potential
+ their framework generalizes to other graph tasks as well
+ they show that even with the attacker knowing how many explanation subgraphs are out there, they cannot realistically find all the subgraphs and claim ownership
+ their method is resistant to removal attacks and data poisoning attacks
+ they were able to show that, using their watermark loss, they were still able to achieve good classification performance

Weakness:
- Same or similar importance of node features can lead to a collision of watermark subgraphs

---

> ### Author Rebuttal · Authors · 2026-03-31
>
> Thank you for your feedback. We appreciate your recognition of our method’s robustness against removal and data poisoning attacks, as well as its ability to generalize across various graph tasks and its potential for real-world deployment. Our detailed responses are provided below.
>
> **W1: If there are node features with similar importance, is it possible to have ambiguous explanations that can be used to claim ownership?**
>
> We emphasize that ownership verification in our framework relies on the **statistical agreement among explanations of T candidate subgraphs** relative to an empirically estimated null distribution. Our method requires **joint agreement across multiple subgraphs**, which is significantly harder to achieve by chance. Moreover, our watermark design **explicitly enforces a margin from the null distribution** by targeting a Matching Index above $\mu_{nat} + z \sigma_{nat}$, thereby increasing separation from naturally occurring agreement.
>
>
> **W2: How to prevent the case when the same or similar node features lead to similar statistical significance, since, as I understood, the importance of each input feature for the GNN’s predictions? Since you mentioned in the footnotes that alignment must simply be “good enough.”**
>
> While it is true that some node features may exhibit similar importance across different subgraphs, our verification criterion is not based on absolute alignment but on **statistical significance relative to an empirical null distribution**. The "good enough" alignment mentioned in the paper refers to achieving a Matching Index that exceeds the natural level of agreement observed among non-watermarked subgraphs, i.e., above $\mu_{nat} + z \sigma_{nat}$. Therefore, even if multiple subgraphs share similar feature-importance patterns, such natural similarity is already accounted for in the null distribution and does not lead to ownership verification. Only **agreement that is significantly stronger than naturally occurring patterns across T subgraphs** is accepted.

---

> > ### Author Rebuttal · Reviewer_Qe8R · 2026-04-01
> >
> > Thank you for the response, I maintain my response.

---

> > > ### Author Response · Authors · 2026-04-01
> > >
> > > Thank you for the positive evaluation. We appreciate the time and effort the reviewer dedicated to providing constructive feedback.

---

### Official Review · Reviewer_BSde · 2026-03-10

**Soundness:** 2
**Presentation:** 2
**Significance:** 3
**Originality:** 2
**Overall Recommendation:** 4
**Confidence:** 3

**Summary:**

The paper introduces a new watermarking method for Graph Neural Networks (GNNs). In particular, the method works in a black-box verification setting for the node prediction task. They use GGN explanations as the main concept behind the watermarking scheme. Finally, they show the effectiveness of the method, as well as robustness and undetectability.

**Compliance With Llm Reviewing Policy:**

Affirmed.

**Final Justification:**

The rebuttal addressed my main concerns (lack of NP-hardness proof and lack of comparison to other baselines), and I raised my score.

**Key Questions For Authors:**

- Can you compare the experimental results with some baselines?
- Can you include a formal proof for NP-hardness?
- Can you provide a working code implementation?

**Limitations:**

yes

**Strengths And Weaknesses:**

Strengths:
- the paper addresses an important subject - watermarking GNNs, and fills the gap created by the lack of such a method
- they use a correct and reflective threat model of real-world scenarios
- the method works nicely, each component is designed and useful
- the results show that the method is effective as well as robust to many attacks and undetectable
- they provide a nice time complexity analysis
- the method shows to be working with multiple GNN architectures
- they show how to extend the method to other graph tasks

Weaknesses:
- the paper does not compare its method to any baselines. Even though they claim to be the only black-box watermarking GNN method (with Xu et al.), I would like to try to adapt other methods to compare numerically. In particular, I would compare the results to Xu et al. If methods such as Xu et al. suffer from issues as described in Related Works, can you show that (compare the methods) experimentally?
- the paper claims to "theoretically prove" that locating the watermark is NP-hard. However, I did not find any theorem/ formal proof for such a strong claim. In particular, in the "undetectability" part of the paper, they claim that the adversary (with access to $G^{tr}$, $T$, and $s$) can, in the worst case, use brute-force or random search. Even though it seems reasonable given the assumptions, there is no detailed explanation/rationale behind that in the paper. Following, even given the assumption of  "brute-force or random search", there is still no formal proof to claim the NP-hardness (at least, in my opinion, the section 4.4 needs to be properly formalized)
- linked code repo is unavailable (expired)
- fairly small number of benchmarks (only 4 datasets)


MInor:
- in 3.1 C is the number of classes, but in 3.2 N is the number of classes
- no std in the scores
- missing a Figure reference in the Appendix
- citing PyTorch paper (Paszke et al.) in reference to structured pruning - is that correct?

---

> ### Author Rebuttal · Authors · 2026-03-31
>
> We sincerely appreciate your comments. We also appreciate your recognition of our method’s effectiveness and robustness against various attacks. Furthermore, we are pleased that you noted its undetectability, low latency, and extensibility across diverse tasks.
>
> Our detailed responses are provided below. Due to the 5000-character limit, we attach empirical results at an anonymous link: https://anonymous.4open.science/r/Rebuttal-D7B4/rebuttal_empirical.pdf
>
> **W1: The paper does not compare its method to any baselines. Even though they claim to be the only black-box watermarking GNN method (with Xu et al.), I would like to try to adapt other methods to compare numerically. In particular, I would compare the results to Xu et al. If methods such as Xu et al. suffer from issues as described in Related Works, can you show that (compare the methods) experimentally?**
>
> We re-implemented Xu et al. (2022) from scratch based on their paper, as no official source code is available. Our implementation faithfully follows their key design choices and optimized hyperparameters (0.15 poisoning rate for GraphSAGE) with a single scalar trigger value drawn uniformly from [0,1] and a fixed target label for all poisoned nodes. We utilize the two datasets used in Xu et al. (2022) (CORA, CiteSeer) and one of the datasets we utilized in our work (PubMed).
>
> Our framework shows similar classification test accuracy as Xu et al. (2022). In terms of verification significance, Xu et al. (2022) failed at seed 3 because the randomized feature value applied to the trigger examples was too small to be distinguished from other regular instances. Our framework shows consistent significance in the verification process in **Table 12** in the link.
>
> As shown in **Table 13** in the link, both methods survive **fine-tuning across** all three datasets.
>
> Beyond the numbers, we highlight a fundamental security distinction. Backdoor watermarking methods embed the mark as an explicit trigger. The trigger is a concrete artifact present in the training data and is detectable via feature outlier analysis or activation clustering. Our method **never modifies the training data**, eliminating this risk entirely.
>
> To illustrate this, inspired by Chen et al. (2018), we show that Xu et al.'s trigger is detectable in raw feature space and can be surgically removed. We run K-Means (k=2) per class on PCA-reduced training features and select the most anomalously separated class via top-1 silhouette score. Within that class, the poison cluster is identified by a const-nonzero heuristic (features with near-zero variance and non-zero mean), directly recovering the trigger positions. Detected nodes are relabeled using model predictions on trigger-zeroed features, then the model is fine-tuned on the corrected data. This forces re-associating the trigger with the correct class. Results across all three datasets (seeds 1–5).
>
> The results can be found in **Table 14** in the link.
>
>
> **W2: Lack of formal proof for NP-hardness claim**
>
> We thank the reviewer for pointing out this. We would like to clarify that we already provide a proof sketch in Section 4.4, where we relate the attacker's search problem to Maximum k-Subset Intersection (MSI). Here we provide a more detailed proof. The NP-hardness claim concerns the attacker's recovery problem of given a collection of candidate subgraphs and their binarized explanations, select exactly ($T$) subgraphs that maximize the MI statistic used in our verifier. Formally, let ($S_i$) denote the signed-support set of nonzero entries of a candidate explanation ($\hat e_i$). Then for any selection ($I$) with ($|I|=T$), we have $\mathrm{MI}(G_i : i \in I) = \left| \bigcap_{i \in I} S_i \right|$.
>
> We reduce MSI to this problem as follows: each MSI subset ($A_i$) is mapped to a candidate subgraph ($G_i$), and each universe element ($u_j$) is mapped to an explanation coordinate ($j$), with ($\hat e_i[j] = +1$) if ($u_j \in A_i$), and (0) otherwise. Under this construction, for any ($k=T$) selected candidates, $\mathrm{MI}(G_i : i \in I) = \left| \bigcap_{i \in I} A_i \right|$. Therefore, solving the attacker's exact search problem would solve MSI, which is NP-hard. This result characterizes the worst-case complexity of the attacker's combinatorial search over candidate subgraphs.
>
> In the revision, we will provide a complete formal proof.
>
>
> **W3: expired code repo**
>
> Thank you for pointing this out. We have restored the visibility of the repository. The code is available in the original repository mentioned in our paper: https://anonymous.4open.science/r/Explanation_Watermarking_GNN-F6C7.
>
> **W4: fairly small number of benchmarks (only 4 datasets)**
>
> Together with results from CORA and CiteSeer above, we use **6 datasets** with broad coverage across scales, sparsity, and domain. The details can be found in **Table 15** in the link.

---

> > ### Author Rebuttal · Reviewer_BSde · 2026-04-02
> >
> > Thank you for the response. I raised my score.

---

> > > ### Author Response · Authors · 2026-04-02
> > >
> > > Thank you for your positive feedback and for increasing your score. We sincerely appreciate your support.

---

### Official Review · Reviewer_Sw26 · 2026-03-12

**Soundness:** 4
**Presentation:** 3
**Significance:** 3
**Originality:** 3
**Overall Recommendation:** 4
**Confidence:** 3

**Summary:**

This paper proposes an explanation-based watermarking method for Graph Neural Networks (GNNs). The method embeds a predefined watermark pattern into the feature attribution vectors of selected subgraphs and verifies ownership via statistical significance testing under a black-box setting. The research addresses an important concept in protecting the intellectual property of deployed GNN models. A central concept explored by the paper is embedding watermarks in the explanation space rather than the prediction space.

**Compliance With Llm Reviewing Policy:**

Affirmed.

**Key Questions For Authors:**

Could the authors include comparisons with existing GNN watermarking methods (e.g., backdoor-based approaches) or fingerprinting methods to better demonstrate the advantages of the proposed method?

**Limitations:**

Please address the questions listed above.

**Strengths And Weaknesses:**

**Strengths:**

S1: The problem is important, as GNN watermarking remains relatively underexplored, especially in black-box settings. The paper considers realistic threat scenarios such as fine-tuning, pruning, and knowledge distillation attacks.

S2: The framework adopts a dual-objective optimization that combines classification loss and a hinge-style watermark loss to align explanations with a watermark pattern. The Matching Index together with a z-test provides a clear statistical verification mechanism.

S3: The authors reduce the problem of locating watermarked subgraphs to the Maximum k-Subset Intersection (MSI) problem and show that it is NP-hard. The paper also approximates the natural MI distribution for statistical testing.

**Weaknesses:**

W1: The novelty appears limited, the method essentially aligns explanation vectors of selected subgraphs with a predefined watermark and verifies ownership via statistical testing. Compared with Explanation as a Watermark (Shao et al., 2024), the main contribution is extending the framework to GNNs with a subgraph-based design and statistical verification.

W2: The stability of the explanation mechanism is not analyzed. Since the watermark is embedded in the explanation space, the robustness of the kernel-based attribution method (ridge regression) under perturbations or different initializations should be examined.

W3: The statistical assumption may be strong. The method assumes binarized explanation entries follow an equal-probability ±1
distribution, while real GNN explanations may exhibit skewed or class-dependent feature importance.

W4: The experimental evaluation lacks several important baselines, such as backdoor-based GNN watermarking methods, fingerprinting approaches (e.g., Waheed et al., 2024), and alternative explanation methods (e.g., GNNExplainer, PGExplainer).

---

> ### Author Rebuttal · Authors · 2026-03-31
>
> We sincerely appreciate your comments, and our detailed responses are provided below. Due to the 5000-character limit, we attach empirical results at an anonymous link: https://anonymous.4open.science/r/Rebuttal-D7B4/rebuttal_empirical.pdf
>
> **W1: In response to question about the novelty of our work and comparison with Shao et al., 2024.**
>
> Shao et al. (2024) cannot be directly applied to the GNN scenario due to two fundamental limitations: (1) retrieving feature attributions from ridge regression on single nodes and their variations does not fully utilize the topological information; (2) lack a proper way to aggregate results for different categories. We incorporate node-wise, kernel-based distances in features and predictions and resolves the fundamental limitations of the application. Importantly, we also introduce a statistical verification mechanism, which is absent in Shao et al.
>
> In an **empirical comparison**, we achieves comparable classification accuracy while producing a statistically significant watermark, whereas Shao et al. (2024) fail the watermark significance test entirely, as shown in our anonymous link in **Table 9**.
>
> In addition, our method is designed for **verification-level stealthiness**, as discussed in Section 4.2. The verification requires only the MI statistic across candidate subgraph explanations, so the watermark pattern w is never disclosed. This is different from methods that require presenting the watermark directly or training a verification classifier.
>
> ---
> **W2: In response to the stability and robustness of the explanation mechanism, in particular under perturbation and with varied initialization.**
>
> We run two experiments that demonstrates the stability and robustness of our method. First, we retrain with seeds 1-5 on data sampling and initialization. Secondly, during inference we inject perturbations on each node feature by independent Gaussian noise with empirical std as 5% / 10% / 20% of the feature values std. In both cases, we observed a small variation in test accuracy and verification p-value. The results can be found in the anonymous link in **Tables 10-11**.
>
> ---
> **W3: In response to the statistical assumption on binarized explanation entries distribution and potential skewed or class-dependent feature importance.**
>
> We clarify that the equal-probability ±1 assumption is used **only in Section 4.3** specifically to analytically predict the null distribution *before training* to determine how many features to watermark ($MI^{tgt}$). This is a design-time convenience that avoids requiring a trained model upfront.
>
> The **actual ownership verification** (Equation 8) uses an entirely empirical null distribution estimated by running ridge regression on a large set of randomly sampled subgraphs from the real graph. This empirical distribution makes **no distributional assumption**. This directly captures whatever skew, class-dependence, or feature imbalance exists in the real GNN's explanations. The equal-probability assumption therefore affects only the *initial sizing* of the watermark, not the correctness of verification.
>
> ---
> **W4: In response to question about baseline such as backdoor-based GNN watermarking methods, fingerprinting approach and alternative explanation methods.**
>
> Regarding **backdoor-based GNN watermarking**, due to space constraints, we **respectfully refer the reviewer to our response to Reviewer BSde, W1**.
>
> As for fingerprinting approaches (e.g., Waheed et al., 2024), we argue that they are not qualified as baselines for our method because they have fundamental limitations that our method resolves:
>
> - For fingerprint methods, **there is no guarantee that independently trained models that share the same data distribution and architecture will not produce similar codes by chance.** Waheed et al. (2024) evaluate on a finite set of independent models but offer no bound that generalizes beyond that set. By contrast, our method embeds the watermark during training; a model never trained with our watermark loss has no systematic alignment with the watermark pattern, and the z-test formally bounds FPR $\leq \alpha_v$ for any non-watermarked model.
>
> - **Fingerprinting decisions rely on empirical thresholds with no statistical grounding.** Waheed et al. (2024) make the ownership decision via majority vote of $C_{\text{sim}}$ outputs at a 50% threshold — chosen arbitrarily which carries no bound on false positive probability. Our z-test provides an unconditional FPR bound, not an empirical one.
>
> As for using other explainers for GNN watermarking, to the best of our knowledge there is no other work that uses GNNExplainer or PGExplainer for GNN watermarking.

---

> > ### Author Rebuttal · Reviewer_Sw26 · 2026-04-02
> >
> > Thank you for the response, I maintain my positive score.

---

> > > ### Author Response · Authors · 2026-04-02
> > >
> > > We sincerely thank the reviewer for the positive feedback and the time and effort dedicated to evaluating our work.

---

### Decision · Program_Chairs · 2026-04-30

**Decision:**

Accept (regular)

**Comment:**

This paper proposes an explanation-based watermarking method for Graph Neural Networks (GNNs).

The reviewers found the studied problem to be important yet underexplored. They praised the realistic threat model and the  black-box assumptions and real-world deployment potential. They found the design of each component justified, and liked that the method can generalize to different GNN architectures and tasks.

Some of the weakness mentioned by the reviewers include the lack of novelty since the method builds on an existing technique for non-graph data (Shao et al., 2024), lack of some baselines, small number of datasets, and a missing proof. The authors adequately addressed these concerns by highlighting the difference to Shao et al., 2024, notably the statistical verification mechanism, they discussed the baselines and datasets, and they provided the missing details for the proof.

The reviewers maintained or increased their score and have a positive overall assessment of the paper after the rebuttal.

Based on the reviews and the rebuttal, I recommend accepting this paper. While the real-world necessity of watermarking GNNs is in my opinion debatable, the technical aspects seem solid and the paper might serve as a starting point for future research in this area.